# Reduced Representation of Deformation Fields for Effective Non-rigid Shape Matching

Ramana Sundararaman[1], Riccardo Marin[2,3], Emanuele Rodolà[3], and Maks Ovsjanikov[1]

[1]LIX, Ecole Polytechnique, IP Paris
[2]University of Tübingen
[3]Sapienza University of Rome

## Abstract

In this work we present a novel approach for computing correspondences between non-rigid objects, by exploiting a reduced representation of deformation fields. Different from existing works that represent deformation fields by training a general-purpose neural network, we advocate for an approximation based on mesh-free methods. By letting the network learn deformation parameters at a sparse set of positions in space (nodes), we reconstruct the continuous deformation field in a closed-form with guaranteed smoothness. With this reduction in degrees of freedom, we show significant improvement in terms of data-efficiency thus enabling limited supervision. Furthermore, our approximation provides direct access to first-order derivatives of deformation fields, which facilitates enforcing desirable regularization effectively. Our resulting model has high expressive power and is able to capture complex deformations. We illustrate its effectiveness through state-of-the-art results across multiple deformable shape matching benchmarks. Our code and data are publicly available at: https://github.com/Sentient07/DeformationBasis.

## 1 Introduction

Shape correspondence is a central problem in computer vision and computer graphics as it facilitates many downstream tasks, such as tracking [1], texture transfer [2] and statistical modeling [3] to name a few. Due to its ubiquitous applicability, a wide range of techniques have been developed over the past several years [4]. While early approaches relied on axiomatic modeling, recent methods follow data-driven techniques based on different input signals [5, 6, 7, 8] within a shape collection.

A key question in this context is the choice of *representation* used to model the non-rigid shape matching problem. Approaches based on *intrinsic* or pose invariant representations have established a gold standard in the context where surfaces are well-defined [9, 10, 11]. Such methods, however, strongly rely on the presence of clean shapes and struggle when acquisition comes from noisy and non-uniform discretization [12]. In contrast, extrinsic techniques which directly operate on Euclidean space ($\mathbb{R}^3$) show strong resilience to artifacts. Unfortunately, this robustness of extrinsic methods often comes at the cost of relying on significant amounts of annotated training data [7, 5]. The main limiting factor arises in the representation of the deformation fields. The standard approach is to use general-purpose MLPs to learn deformation fields that can fit an arbitrary shape deformation [13, 14, 5]. However, given the fact that MLPs are general-purpose networks, they require significant amounts of training data to learn both coarse and fine details [5].

To overcome this limitation, we propose to learn a coarse representation of *deformation parameters* at fixed positions in space called "nodes". By learning a reduced representation of deformation fields, intuitively, we restrict the learning process to global patterns of the input signal. Then, to recover

36th Conference on Neural Information Processing Systems (NeurIPS 2022).

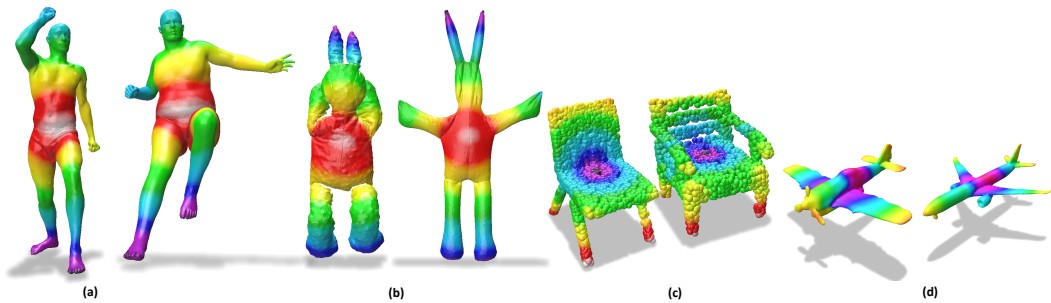

Figure 1: Examples showing generalization of our deformation field representation. Our approach allows to compute correspondences across a wide range of shape categories: (a) human articulation, (b) physically-based deformation from real-world scans, (c) shapes undergoing topological changes, and (d) shapes represented via implicit functions.

finer details, we reconstruct the continuous deformation field function in closed-form using a class of mesh-free approximation techniques [15]. This allows us to scale our approach to arbitrary resolution with guaranteed smoothness and across different object classes as shown in Figure 1.

Apart from being theoretically grounded and simple in practice, our reduced representation has two key advantages. First, it is significantly more data-efficient and can learn to capture complex deformations given only a small number of examples. Secondly, it is more amenable to regularization, since it provides explicit access to first-order derivatives of the deformation field in closed form. This is especially useful in imposing geometric priors such us local rigidity and volume preservation.

Our contributions can be summarized as follows: (a) We propose to learn a compact representation of deformation parameters, that is data-efficient, resolution agnostic, and facilitates regularization through direct access to deformation gradients (b) We show an efficient way of incorporating desirable regularization to promote a well-structured deformation space. (c) Through extensive experiments across real-world and synthetic datasets, we demonstrate the generalization ability of our method over different down-stream applications such as non-rigid shape matching, registration, unsupervised part segmentation and interpolation.

## 2 Related Works

### 2.1 Non-rigid shape Correspondence and Registration

Shape correspondence is a very well-studied area of computer vision and computer graphics and we refer interested readers to the recent survey [4] for a comprehensive overview. Notable axiomatic approaches in this category are based on the functional maps paradigm [9, 16, 17, 18], that aims to compute a near-isometric mapping by estimating a linear transformation between functions represented in a reduced basis. This framework has been successfully adapted by learning techniques [6, 19, 20, 7, 10] which demonstrate near-perfect accuracy [11] on several shape correspondence benchmarks. However, these approaches can be prone to errors in the presence of noisy point clouds or significant acquisition artefacts. Although registration-based techniques [21, 22, 23] present a relatively more robust option, they are often based on human-centric priors or require significant training data.

### 2.2 Template-based and Template-free Methods

Deforming a template shape to match a target geometry is a long-standing and well studied problem [24, 25, 26]. Such a template can be a polygonal mesh [27, 28], possibly parameterised [29, 30, 31] or an unordered point-set [32, 33, 34, 35, 36] or implicitly defined through zero-level set of a Neural Field [13, 14, 37]. In the recent years, learning based model-free deformation techniques [32, 33, 23] have emerged as a viable option for registration and correspondence tasks given copious amount of training data [32, 38]. Among them, the closest to our approach is 3D-CODED [32] which learns deformation fields through point-wise MLPs. However, since this

approach fits a general-purpose MLP and treats each point on the shape equally likely, it requires abundant training data to achieve optimal performance.

## 2.3 Deformation Field Representation and Shape Interpolation

Deformation between a pair of shapes can be represented as a simple displacement field at every vertex. However, such a representation can be unnecessarily complex, and costly to optimize. As a result, several alternatives have been proposed. The most prominent ways to parameterize the space of deformations include handle-based [39, 40, 41, 42] or cage-based [43, 44, 45, 46] representations (see also [47, 48, 49] for an overview). More recently, a common approach is to construct a reduced representation via a learned latent embedding [50, 51, 52, 53].

**Deformation Field regularization**    Several geometric constraints have been proposed with the aim of preserving desirable properties of the shape by the deformation field, including imposing elasticity [54, 55, 56] and volume preservation [57, 58, 40, 59, 60]. Recently, these constraints have been successfully adapted by data-driven methods [61, 51, 62, 37, 63] and more relevantly through the differential of the map [51, 63, 63]. Distinct from such approaches, our approximation via mesh-free method enables evaluating this map differential at fixed points in a closed-form, which significantly simplifies the deformation field regularization without additional computational overhead.

**Shape Interpolation**    Shape interpolation refers to time-parameterized deformation, where a source shape is continuously deformed to a target shape. Our work is related to efforts which aim to enforce intermediate shapes to preserve certain intrinsic properties [52, 53, 61, 64, 65, 51]. Among them closest to our approach is LIMP [52] which disentangles the latent space based on style and pose to preserve geodesic distance. In contrast, our approach does not require such a priori information, which can be costly in terms of annotation efforts.

## 2.4 Reduced representations and Approximations

In this work we use mesh-free function approximation method [66, 67, 15], to approximate deformation fields. Mesh-free methods have been successfully adapted in Smoothed Particle Hydrodynamics (SPH) modeling [68, 69], image processing [39], animation [40, 70, 71] and more recently in a data-driven framework [41]. Differently from [41], instead of learning the weights of the least squares function, we instead learn deformation values at nodes and demonstrate our method to be applicable in wide-range of downstream tasks. Alternatively, Eisenberger *et al.* [60] have proposed to use a compact representation of deformation fields using the first $k$ eigenfunctions of the Laplace Beltrami Operator (LBO). While their approach provides volume preserving deformation, it does not facilitate other regularizations such as as-rigid-as-possible deformation fields without requiring correspondence at inference time [72].

# 3   Motivation, Background and Notation

## 3.1   Motivation:

Parametric models such as SMPL [29] have been tremendously useful over the recent years in digitizing and processing human models. This success can largely be attributed to their expressive power, allowing to generate a wide range of styles and poses using a small fixed set of deformation parameters. While this efficacy with such a compact representation is remarkable, it also raises an inspiring question: what is the optimal amount of *learnable* parameters necessary to represent general deformations? Today, general-purpose MLPs form the conventional way of representing deformation fields due to their simplicity and potential of being universal functional approximators [73, 74]. Unfortunately, the generic power of MLPs also comes at a cost of copious training efforts [32, 23]. Furthermore, representing a deformation field using a neural network makes access to certain quantities such Jacobian matrices of deformation fields cumbersome. For these reason, we propose to learn a reduced set of deformation parameters from which we *approximate* the deformation field function using a mesh-free method.

## 3.2 Mesh-free Approximation

Mesh-free methods are a class of approximation techniques which constructs a continuous function based on independent, potentially sparse and irregular observations. Assume that our domain of interest $\mathbb{R}^3$ is equipped with $K$ fixed points $\mathbf{q}_i \in \mathbb{R}^3$ along with some observations $u_i$ at $\mathbf{q}_i$ and a choice of a polynomial basis $p(\cdot)$. We refer to fixed points $\mathbf{q}_i$ as "nodes" (or, alternatively, "deformation nodes"). Our main goal is to construct a continuous approximation of some real-valued function $u(.)$ in some subdomain $\Omega \subset \mathbb{R}^3$ of interest. We let $\mathbf{x} \in \Omega$ to be an arbitrary point in our region of interest. The key idea behind this approximation is to use a *local weighted least-squares* fitting (also referred to as Moving Least Squares) approach [15]. Specifically, we first build a compactly supported weighting function $w_i(\mathbf{x})$ in the neighborhood of $\mathbf{q}_i$, via:

$$
w_i(\mathbf{x}) = \begin{cases} \left(1 - \dfrac{||\mathbf{x} - \mathbf{q}_i||_2^2}{r_i}\right)^3, & \text{if } ||\mathbf{x} - \mathbf{q}_i||_2^2 \leq r_i \\ 0, & \text{otherwise} \end{cases} \tag{1}
$$

The compactness of this weighting function is useful in preserving the local characteristics of approximation. From this, a *Shape Function* $\Phi_i$ associated with each node $i$, is constructed as:

$$
\Phi_i(\mathbf{x}) = p^T(\mathbf{x})[M(\mathbf{x})]^{-1} w_i(\mathbf{x}) p(\mathbf{q}_i). \tag{2}
$$

Here $M(\mathbf{x})$ is the Moment Matrix associated with the approximation, and defined as:

$$
M(\mathbf{x}) = \sum_{i=1}^{K} w_i(\mathbf{x}) p(\mathbf{q}_i) p^T(\mathbf{q}_i)
$$

The shape function $\Phi_i$ is a continuous function that describes how each node $\mathbf{q}_i$ influences the approximation of $u(.)$ across points $\mathbf{x} \in \Omega$. Jointly the the set of $\Phi_i$'s enable the reconstruction of arbitrary functions up to $n^{th}$ order consistency [67], where, $n$ is the order of the polynomial $p(\cdot)$. Specifically, a smooth local approximation of $u(\mathbf{x})$ is given as:

$$
u(\mathbf{x}) = \sum_{i=1}^{K} \Phi_i(\mathbf{x}) u_i \tag{3}
$$

As the construction of $\Phi$ involves computing $M^{-1}$ (c.f Eq. (2)), it is a sufficient condition for each point $\mathbf{x}$ to be compactly supported by 4 non-planar nodes $\mathbf{q}_i$ for $M$ to be non-singular. It is important to note that Eq. (3) is approximating and *not interpolating*, i.e $u_i \neq u(\mathbf{q}_i)$. For instance, owing to the compact nature of $w(\mathbf{x})$, it is possible that $u(\mathbf{q}_i)$ is undefined if $\mathbf{q}_i \notin \Omega$. For this reason, we sample the nodes a priori to have a well-supported domain $\Omega \subseteq \mathbb{R}^3$ where $u(\mathbf{x})$ is well-defined.

Furthermore, an important advantage of using mesh-free approximations comes from an exact analytical expression for the gradient function of $u(\mathbf{x})$. To the scope of our current discussion, considering $u(\mathbf{x})$ to be the approximation of deformation field function, the Jacobian of this deformation field only depends on evaluation point and is independent of observed deformation parameters $u_i$ ,

$$
\mathbb{J} = \nabla_{x,y,z} u(\mathbf{x}) = \sum_{i=1}^{K} \left[\frac{\partial \Phi_i(\mathbf{x})}{\partial x}, \frac{\partial \Phi_i(\mathbf{x})}{\partial y}, \frac{\partial \Phi_i(\mathbf{x})}{\partial z}\right]^T u_i \tag{4}
$$

This nice property helps us characterize the deformation fields with desired first-order regularization in an efficient manner. We refer interested readers to [67, 66] for a detailed summary.

## 3.3 Notation:

As our training set, we consider a collection of shapes $\{\mathcal{S}_1 \ldots \mathcal{S}_N\}$ with ground truth correspondences $\Pi_{\mathcal{S}_l \mathcal{S}_j}$ between them. Shapes can be represented as triangular meshes $\mathcal{S}_j := \{\mathcal{V}, \mathcal{E}\}$ or simply unordered sets of points (point clouds) $\mathcal{S}_j := \{\mathcal{V}\}$. We pick one shape from the collection as a template $\mathcal{T}$, and let $[\mathcal{T}]$ be the volume enclosed by the boundary $\partial \mathcal{T}$. We refer to $\mathcal{Q} \in \mathbb{R}^{K \times 3}$ as

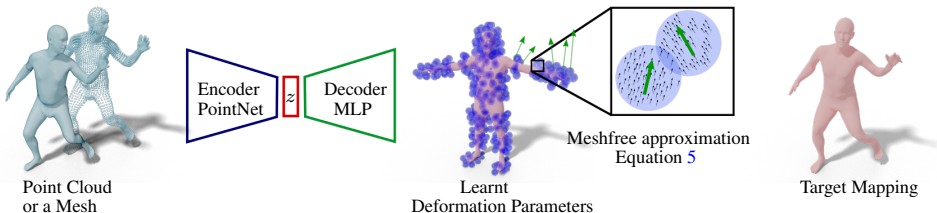

Figure 2: Overview of our approach. First, we learn the deformation parameters at nodes using an Auto-Encoder. Then, we use mesh-free approximation to obtain a continuous deformation mapping.

*nodal positions*, which are *K fixed* points in space sampled from the template volume $[\mathcal{T}]$. We let $\mathcal{D}(\cdot) : \mathbb{R}^3 \rightarrow \mathbb{R}^3$ be the *deformation mapping*, which, intuitively maps points in the deformation volume to points on target shapes. We refer to $U_j$ as the nodal deformation parameters corresponding to the $j^{\text{th}}$ shape and analogously define $D_j(\cdot)$. Each node $q_i \in \mathcal{Q}$ has a support radius $r_i$ and associated deformation parameter $u_i$. We use lower-case notation $u_{i,j}$ to refer to the value of the deformation field at node $\mathbf{q}_i$ corresponding to shape $j$. We denote $\mathbf{x} \in \Omega \subset \mathbb{R}^3$ as points in space which are supported by at least four non-planar nodes. We refer to $U_j(\mathbf{x})$ as the continuous approximation of the deformation field, constructed from deformation parameters $U_j$ using Eq. (3). We re-iterate that $u_{i,j} \neq U_j(\mathcal{Q}_i)$. The relation between a deformation *field* and a deformation *mapping* is given by $\mathcal{D}_j(\mathbf{x}) := \mathbf{x} + U_j(\mathbf{x})$. For the sake of consistency, we index nodes using $i$, shape collection using $j, l$ and points within shape using $k$.

# 4 Method: Learning Nodal Deformation-Field

**Overview.** Our network is based on a PointNet [75] auto-encoder as shown in Figure 2. Our network $\mathcal{F}_\theta(\cdot)$ predicts nodal deformation parameters $U_j$ for each training shape $\mathcal{S}_j$, i.e $U_j = \mathcal{F}_\theta(\mathcal{S}_j)$ where $U_j \in \mathbb{R}^{K \times 3}$. As mentioned before, the nodes $\mathcal{Q}$ are fixed a priori. From the predicted $U_j$, we can compute the shape-specific deformation mapping $\mathcal{D}_j(x)$ and its Jacobian $\mathbb{J}_j$, via:

$$\mathcal{D}_j(\mathbf{x}) = \mathbf{x} + \sum_{i=1}^{K} \Phi_i(\mathbf{x}) u_{i,j}$$
$$\mathbb{J}_j = \mathbf{I} + \nabla_{x,y,z} U_j(\mathbf{x}) \tag{5}$$

Where, $\nabla_{x,y,z} U_j(\mathbf{x})$ is given in Equation 4.

## 4.1 Training

Intuitively, we would like to train a network so that $S_j \approx \{D_j(\mathbf{x}) | \mathbf{x} \in \mathcal{T}\}$, subject to appropriate regularization. Although $\mathcal{D}_j(\mathbf{x})$ can be approximated at an arbitrary $\mathbf{x}$, which is supported by four non-planar nodes, we restrict ourselves to $\mathbf{x} \in \mathcal{T}$ for the ease of learning. As $\mathcal{F}_\theta(\cdot)$ represents an auto-encoder, it can be decomposed as $\mathcal{F}_\theta(\mathcal{S}_j) = Dec(Enc(\mathcal{S}_j)) = Dec(Z_j)$ where $Z_j$ denotes the latent embedding. Leveraging this fact, we provide a novel way to promote plausible latent deformation spaces $Z_j$ by enforcing first-order constraints over the *intermediate shapes* as well. The overall optimization objective of our network is given as:

$$\mathcal{L}_{net} = \lambda_1 \mathcal{L}_{cor} + \lambda_2 \mathcal{L}_{vol} + \lambda_3 \mathcal{L}_{arap} + \lambda_4 \mathcal{L}_Z \tag{6}$$

For the unsupervised case, we replace $\mathcal{L}_{cor}$ with $\mathcal{L}_{CD}$ which denotes the Chamfer's distance.

**Correspondence Loss** Given a set of $\mathcal{C}$ of corresponding points $\{\mathbf{x}_l, \mathbf{x}_k\}$, where $\mathbf{x}_l \in \mathcal{T}$, $\mathbf{x}_k \in \mathcal{S}_j$ our correspondence loss is given as

$$\mathcal{L}_{cor} = \sum_{j=1}^{N} \sum_{(\mathbf{x}_k, \mathbf{x}_l)}^{|\mathcal{C}|} \left\| \mathcal{D}_j(\mathbf{x}_l) - \mathbf{x}_k^j \right\|_2^2 \tag{7}$$

Where $x_k^j$ denotes the $k^{th}$ point in $j^{th}$ shape.

**Volume Preserving Field**  A deformation field is volume preserving iff its Jacobian has unit determinant over the entire shape. Consequently, our local volume preservation regularization is given as follows:

$$\mathcal{L}_{vol} = \sum_{j=1}^{N} \sum_{i=1}^{K} |\det(\mathbb{J}_j(\mathbf{q}_i)) - 1|_2^2 \tag{8}$$

We empirically observe poor convergence when this objective is enforced over the entire shape due to its stringent nature. This is because not all deformations are strictly volume preserving. Thus, we restrict this regularization only at nodes.

**As Rigid As Possible (ARAP) Deformation**  Since rigid motions preserve pairwise distances, a deformation field associated with such a transformation is characterized by an orthonormal Jacobian matrix. Thus, in order to promote locally rigid deformation field at the deformation nodes, we define our ARAP regularization as:

$$\mathcal{L}_{arap} = \sum_{j=1}^{N} \sum_{i=1}^{K} \left\| \mathbb{J}_j^T(\mathbf{q}_i) \mathbb{J}_j(\mathbf{q}_i) - \mathbf{I} \right\|_F^2 \tag{9}$$

**Structuring Latent Deformation Space**  A well-known advantage of an auto-encoder architecture is the construction of the latent space, where each shape has an embedding $Z_j \in \mathbb{R}^D$. Then, a parameterized path in this latent space between two shapes $Z_{l,j}(\alpha) = \alpha Z_l + (1 - \alpha) Z_j$ continuously deforms $\mathcal{S}_j$ to $\mathcal{S}_l$ with rate of change controlled by $\alpha$. This allows constructing a sequence of shapes, often referred to as interpolated shapes. Since each $Dec(Z_j) = U_j$, we can further require our network $\mathcal{F}_\theta$ to produce a plausible deformation between *each pair of training shapes*. To that end, we introduce our latent smoothness loss as follows:

$$\mathcal{L}_Z = \sum_{l \neq j}^{|\mathcal{S}|} \mathcal{L}_{arap} \left( \dec((1 - \alpha)\mathbf{z}_j + \alpha \mathbf{z}_l) \right) + \sum_{l \neq j}^{|\mathcal{S}|} \mathcal{L}_{vol} \left( \dec((1 - \alpha)\mathbf{z}_j + \alpha \mathbf{z}_l) \right) \tag{10}$$

Where $\mathcal{L}_{arap}, \mathcal{L}_{vol}$ are defined in Eqs. (8), (9) and $\alpha \in (0, 1)$ are sampled randomly.

## 4.2 Inference

At test-time, given a pair of unseen shapes $(\mathcal{X}, \mathcal{Y})$ we follow a three-step procedure to obtain the correspondence $\Pi_{\mathcal{X}\mathcal{Y}}$. First, we separately reconstruct $(\mathcal{D}_\mathcal{X}, \mathcal{D}_\mathcal{Y})$ by deforming the fixed template $\mathcal{T}$. Second, we enhance the respective reconstructions by optimizing the latent vector $Z$ independently for shapes $(\mathcal{X}, \mathcal{Y})$. The objective for this optimization is to minimize the bi-directional Chamfer Distance [32] while *also* enforcing first-order constraints as follows:

$$Z = \underset{Z}{\mathrm{argmin}} \, \Lambda_1 \mathcal{L}_{\mathcal{CD}} + \Lambda_2 \mathcal{L}_{arap} + \Lambda_3 \mathcal{L}_{vol}. \tag{11}$$

As $\mathcal{D}_\mathcal{X}$ is the reconstruction of $\mathcal{X}$, the correspondence between $\mathcal{D}_\mathcal{X}, \mathcal{X}$ can be computed via a simple nearest neighbor search in 3D (analogously for $\mathcal{Y}$). Since $\mathcal{D}_\mathcal{X}, \mathcal{D}_\mathcal{Y}$ are deformed versions of a template they enjoy a natural correspondence (by vertex ordering). Finally, the correspondence between $(\mathcal{X}, \mathcal{Y})$ is a composition of two nearest neighbour searches $\Pi_{\mathcal{X}\mathcal{Y}} == (\mathrm{NN}(\mathcal{D}_\mathcal{X}, \mathcal{X}), \mathrm{NN}(\mathcal{D}_\mathcal{Y}, \mathcal{Y}))$.

## 4.3 Extending sparse to dense Correspondence

An added advantage of our representation is the ability to retrieve dense shape correspondence between a shape pair, given a few sparse key-point correspondences $(\mathbf{x}_l, \mathbf{y}_k), \forall \mathbf{x}_l \in \mathcal{X}, \forall \mathbf{y}_k \in \mathcal{Y}$. First, we estimate the deformation parameter $u_i$ at the nodes by solving an optimization:

$$u_i = \underset{u_i}{\mathrm{argmin}} \, \lambda_1 \sum_{\forall (\mathbf{x}_l, \mathbf{y}_k)} \|\mathcal{D}_\mathcal{X}(\mathbf{x}_l) - \mathbf{y}_k\|_2^2 + \lambda_2 \left\| \mathbb{J}_\mathcal{X}^T \mathbb{J}_\mathcal{X} - I \right\|_F^2 + \lambda_3 |\det(\mathbb{J}_\mathcal{X}) - 1|_2^2 \tag{12}$$

Then, a dense mapping can be computed by approximating the deformation field (c.f Equation 5).

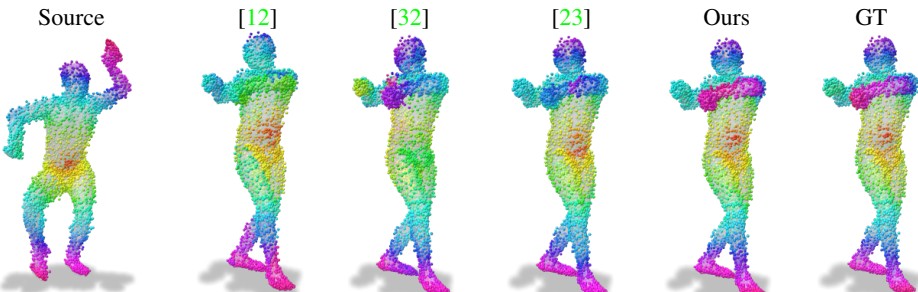

| Source | [12] | [32] | [23] | Ours | GT |

Figure 3: Color-coded correspondences on the SCAPE (PC+N) dataset. "Twist" is a challenging articulation as a wrong deformation can lead to large geodesic error (see Cheese-Pull effect in [76], Figure 11). We hypothesize that our approach, which learns a "global" sense of the articulation, does not suffer from such artefacts since the fine (local) details are computed in closed form.

| | Method | | Correspondence Error | | |
|---|---|---|---|---|---|
| Type | Name | #Tr data | SHREC'19 | FAUST(NI) | SCAPE(PC+N) |
| Spectral | GeoFMap [7] | 1.7 | 11.2 | 20.1 | 27.7 |
| Pair-wise | Diff-FMap [12] | 1.0 | 15.1 | 5.4 | 26.0 |
| | CorrNet3D [8] | 15.0 | 9.6 | 25.9 | 38.0 |
| Template based | 3D-CODED [5] | 23.0 | 10.3 | 7.0 | 18.7 |
| | TransMatch [23] | 1.0 | 6.1 | 6.5 | 17.1 |
| | Ours | **0.1** | **4.8** | **5.3** | **6.6** |

Table 1: We report correspondence error as geodesic distortion (in cm) scaled by square root of shape area. #Tr data denotes number of training shapes scaled by $10^{-4}$.

## 4.4 Implementation details

**Analytical Gradients and Timing advantages**   We leverage the advantage of inexpensive access to Jacobians as mentioned in Equation 4. Because our evaluation points are known a priori, due to the use of a fixed template $\mathcal{T}$, the matrix $\mathbb{J}$ can be pre-computed and re-used at training and evaluation. In practice, we observe a $10\times$ speed-up at training time when enforcing our first-order constraints and a $350\times$ speed-up incorporating the latent constraints (c.f. Eqn 10). We provide more timing details in the supplementary.

**Node Sampling:**   Since the deformation field at a point is determined by the nodes within the radius, it is important to limit the influence of a node which is close in a Euclidean sense but geodesically far. For instance, it is counter-intuitive to have a node in the trunk of the human influencing the deformation of a point in the arm. Bearing this in mind, our node sampling strategy is divided into three main steps. First, we construct a dense sampling of points in the volume and around the boundary of the template $\partial\mathcal{T}$. Second, we use rejection sampling to *exclude* a node that exerts its influence in semantically different regions [29]. Finally, we perform Farthest Point Sampling (FPS) until each surface point is covered by 4 non-planar nodes. We emphasize that this step is performed only on the template shape and using SMPL [29] segments is *one of many* possible ways to perform segmentation. An in-depth ablation study is provided in the supplementary material.

## 5  Experiments

The reduced representation for deformation field which we have discussed so far is conducive to produce naturally smooth deformation while significantly reducing the amount of supervision needed to facilitate learning. We empirically show the efficacy of our proposed representation of deformation fields across four main tasks, namely Non-rigid 3D shape correspondence, Shape registration, Unsupervised part segmentation and Shape interpolation.

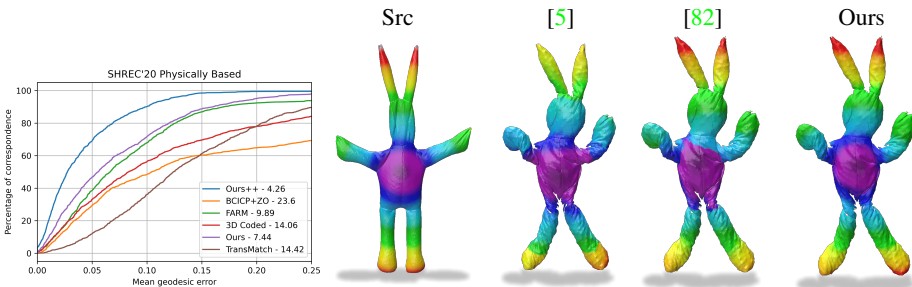

Figure 4: Quantitative and qualitative results on SHREC'20. Our approach predicts smooth correspondences across highly-granular surface-level deformation.

## 5.1 Shape Correspondence

We consider three challenging benchmarks, namely, SHREC'19, FAUST (PC), SCAPE (PC+N). SHREC'19 [77] is a standardised benchmark consisting of 430 evaluation pairs with significant variations in mesh resolution and connectivity. FAUST (PC) denotes a more recent Non-Isometric Point Cloud variant [12] of the FAUST dataset consisting of 1000 points with large variance in point sampling density. Third, we evaluate on a variant of the recent SCAPE-Remesh dataset [78] consisting of 20 shapes of the same human in 20 distinct poses. We further augment the challenge by adding random Gaussian noise and refer to as SCAPE (PC+N). We evaluate correspondence error following the Princeton benchmark protocol [79]. We train our method on a subset of 1000 SURREAL shapes [80] for 1000 epochs with data-augmentation along Y-axis.

**Baselines**    We compare our method against data-driven correspondence methods broadly classified into Spectral, Pairwise and Template based. We use GeoFMap [7] with the more robust feature extractor Diffusion-Net [11] as our spectral baseline, Diff-FMaps [12] and CorrNet3D [8] as our pairwise baseline. For our template based baselines, we use 3D-CODED [32] and TransMatch [32]. For the evaluation of baselines on our proposed SCAPE (PC+N), we use the author-provided pre-trained models, and apply consistent preprocessing to the input shapes across all methods.

**Discussion**    Our approach consistently outperforms baselines as summarised in Table 1. While our quantitative correspondence results are persuasive, it is remarkable to note that our method requires an order of magnitude less training data in comparison to competing methods. This supports our premise that characterizing typical deformations requires far fewer parameters than what is leveraged by existing data-driven methods. We show qualitative correspondence results through color transfer for a challenging pair with "twisted" motion in Figure 3.

## 5.2 Shape Registration

Shape registration is a special case of correspondence, where our goal is to find an optimal deformation between the scan and a fixed template. For this, we consider the recent SHREC'20 benchmark [81], consisting of 11 partial scans of stuffed toy rabbits to be registered to a single scan. This benchmark is particularly challenging due to granulated surface deformation, scanning artefacts, and limited data and supervision.

**Experiment**    We split this dataset into 7 training shapes and 4 shapes for evaluation. Shapes in our test set are made of "chickpea" material, which exhibits the largest magnitude of granular surface deformation. We compare our method with 4 baselines namely, FARM [21], BCICP [78]+ZoomOut [83], 3D-CODED [32] and TransMatch [23]. Since the two data-driven baslines are not designed for training with key-point supervision, we use Equation 12 to generate dense-ground truth for training. In fairness, we report two variants of our method trained - one trained with key-point and the other with dense supervision denoted as "Ours" and "Ours++" respectively. We stress that this additional supervision is used only at training time while we maintain the test set to be fully-blind. We summarize our quantitative and qualitative results in Figure 4. It is remarkable that our approach outperforms

| | Plane | | Table | |
|---|---|---|---|---|
| | CD (x1e4) | IoU (%) | CD (x1e4) | IoU (%) |
| DIT [14] | 24.6 | 69.1 | 26.7 | 68.9 |
| DIF [13] | 15.0 | 78.0 | 11.2 | 79.3 |
| Ours (w/o Con) | **0.5** | 88.8 | **2.6** | 88.9 |
| Ours | 0.6 | **89.3** | 3.0 | **90.0** |

Source    DIT    DIF    Ours (w/o Con)    Ours

Figure 5: Ours (w/o Con) is a variant of our approach without any deformation constraints. Both of our variants show a significant improvement over baseline that models volumetric deformation fields. Qualitative result demonstrates color coded segmentation transfer across significant shape variability.

axiomatic and competing data-driven baselines by at least a two-fold margin. Importantly, despite our network sharing the same encoder [75] as 3D-CODED [32], there is a striking difference in performance. We attribute this to our well-regularized deformation space.

## 5.3 Unsupervised Segmentation Transfer

In this section, we demonstrate the generalization ability of our approach to model deformation between shapes with considerable topological differences. To that end, we consider the task of part-level segmentation over point clouds consisting of table and plane categories from ShapeNet [84] dataset. Apart from topological differences and large structural variance, the absence of ground truth annotations exacerbates the challenge. In this setting, we compare our method with two Deep Implicit networks, namely DIF-Net [13] and DIT [14], which model a volumetric deformation field between a learned template and training shapes. Our choice of baseline endows us with a fair ground of comparison between the two representations of the deformation field - MLP-based and ours.

**Experiment and Discussion** We train our approach using the unsupervised loss mentioned in Section 4.1 over 1000 objects sampled at random from each category. We consider 190 evaluation pairs per-category from the prescribed validation set and measure the segmentation accuracy by the IoU metric [84]. In addition, we also measure the bi-directional Chamfer's distance of reconstructed geometries. We summarize our quantitative observation along with a qualitative example in Figure 5. We remark that while deformation fields between aforementioned categories are not strictly volume-preserving, we still observe a noticeable improvement over the baseline. This is because our deformation priors help in *structuring* the space of deformations, which explicitly avoids degeneracy such as collapsing shape parts. This remark is corroborated by a lower (preferred) Chamfer distance while there is *a decline* in the accuracy when no regularization is applied.

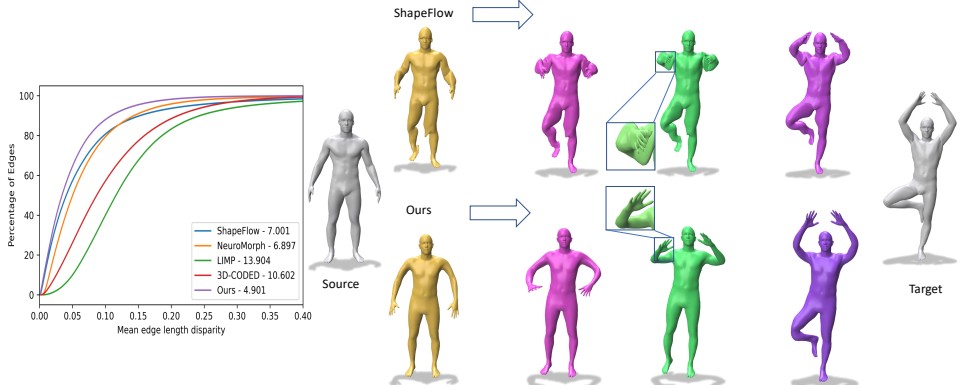

Figure 6: Quantitative and qualitative comparison of interpolation. While ShapeFlow [51] enforces volume preservation prior, its latent deformation space are not distortion-free. By better structuring the latent-space (c.f Eqn 10), our sampled intermediate shapes are near distortion-free.

## 5.4 Shape Interpolation

A notable characteristic of a well-structured latent space is the ability to produce plausible intermediate shapes given a source and a target. This task is commonly referred to as shape interpolation. Since there exists no canonical path, interpolation sequences are gauged by the extent to which intrinsic metrics are preserved, in particular isometric distortion [85]. For this setting, we consider the FAUST [3] dataset, where, we train our method on the first 80 shapes and evaluate over the last 20 shape pairs. We compare our method against four baselines namely 3D-CODED [5], NeuroMorph [61], LIMP [52] and ShapeFlow [51]. Since LIMP employs a fixed-size decoder and NeuroMorph uses a separate interpolation module involving an explicit computation of correspondence matrix, both of these approaches are limited by shape resolution. On the other hand, our approach is resolution agnostic and outperforms the baselines by a discernible margin as summarized in Figure 6. This improvement over the baseline is due to the incorporation of our latent deformation priors in a computationally feasible manner, which we will be justified through an extensive ablation study in the supplementary.

### Additional Results

In addition to the results shown above, we also present qualitative correspondence results between neural implicit fields and real-world data in the supplementary. More specifically, in Section 5 of the supplementary, we show qualitative interpolation and correspondence results between implicitly defined surfaces. Then, in Section 6.1 of supplementary, we show qualitative correspondence results in the form of texture transfer between pair of shapes from the CMU-Panoptic dataset [86] consisting of point clouds acquired from from *Kinect RGB-D sensor*. Finally, in Section 6.2, we also show the versatility of our representation in modelling deformation field between shapes that have more freedom regarding such as meshes of the human heart [87].

## 6 Conclusion, Limitations and Future Work

We presented an effective representation of deformation fields, which allows learning a *reduced set* of shape-specific deformation parameters while constructing the continuous deformation field using mesh-free approximation. A key observation behind our method is that in many settings, the space of realistic deformations is well-constrained and expressed with a small set of parameters. To that end, we demonstrated that our approach can achieve significant improvement upon existing baselines across challenging downstream applications and remarkably reduce the dependence on training data. Moreover, this representation also endowed us with access to first-order derivatives in closed form, thereby facilitating the use of strong first-order regularization. Our approach still has some limitations and leads to possible exciting future work. Firstly, while our approach produces a smooth deformation field in principle, there is no guarantee of bijectivity or invertability. Second, instead of fixed nodal positions, optimizing with respect to our approximation function would also be an interesting direction to study.

**Acknowledgments:** Parts of this work were supported by the ERC Starting Grants No. 758800 (EXPROTEA) and No. 802554 (SPECGEO), the ANR AI Chair AIGRETTE, an Alexander von Humboldt Foundation Research Fellowship. We thank Robin Magnet and Gautam Pai for their feedback in improving our manuscript.

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
