# Reduced Representation of Deformation Fields for Effective Non-rigid Shape Matching: Supplementary Material

Ramana Sundararaman[1], Riccardo Marin[2,3], Emanuele Rodola[2], and Maks Ovsjanikov[1]

[1]LIX, Ecole Polytechnique, IP Paris
[2]Sapienza University of Rome
[3]University of Tübingen

## Abstract

In Section 1 we provide the implementation details of our novel deformation field representation, followed by a quantitative comparison with potential alternative representations in Section 2. Then, we perform an extensive ablation study in Section 3 to justify the need for regularization and our design choices. In Section 4, we quantitatively show the reduced need for supervision of our approach by comparing with relevant baselines. In Section 5, we further highlight the generalization ability of our reduced representation in establishing high-quality correspondences between learnt implicitly defined surfaces. Finally in Section 6, we demonstrate the robustness of our approach in estimating correspondence between real-world data acquired from RGB-D sensor and scans of human hearts.

## 1 Implementation Details

### 1.1 Pre-Processing

We scale all training shapes to fit into a unit sphere including our template. Then, we sample nodes from within the volume defined by the template if our shape collection is a mesh. In case of point cloud, we simply augment by adding random Gaussian noise. Once sampled, we fix the positions of nodes. Finally, we pre-compute $\Phi$ and $\nabla_{x,y,x}\Phi$ at each evaluation points. Please note this pre-computation is performed only to accelerate training when evaluation points are known and it is not a strict requirement.

### 1.2 Closed-form expression for deformation field gradient

We show the pre-computation of $\nabla_{x,y,x}\Phi$ in this subsection. From Equation 4 of the main paper, the Jacobian of the deformation field is given as,

$$
\mathbb{J} = \frac{\partial \mathbf{u}(\mathbf{x})}{\partial \mathbf{x}_{(d)}} = \sum_{i=1}^{K} \frac{\partial \Phi_i(\mathbf{x})}{\partial \mathbf{x}_{(d)}} u_i \quad \text{where,} \ \ \mathbf{x}_{(d)} = [x, y, z]^T
$$

Expanding $\frac{\partial \Phi_i(\mathbf{x})}{\partial \mathbf{x}_{(d)}}$,

$$\frac{\partial \Phi_i(\mathbf{x})}{\partial \mathbf{x}_{(d)}} = \frac{\partial \left(p^T(\mathbf{x})[M(\mathbf{x})]^{-1}w_i(\mathbf{x})p(\mathbf{q}_i)\right)}{\partial \mathbf{x}_{(d)}}$$

$$= \frac{\partial p^T(\mathbf{x})}{\partial \mathbf{x}_{(d)}}[M(\mathbf{x})]^{-1}w_i(\mathbf{x})p(\mathbf{q}_i) + p^T(\mathbf{x})\frac{\partial [M(\mathbf{x})]^{-1}}{\partial \mathbf{x}_{(d)}}w_i(\mathbf{x})p(\mathbf{q}_i) + p^T(\mathbf{x})[M(\mathbf{x})]^{-1}\frac{\partial w_i(\mathbf{x})}{\partial \mathbf{x}_{(d)}}p(\mathbf{q}_i)$$

Using a $1^{st}$ order polynomial basis $p(\mathbf{x}) = \begin{bmatrix} 1 \ x \ y \ z \end{bmatrix}^T$ and the fact that $\frac{\partial [M(\mathbf{x})]^{-1}}{\partial \mathbf{x}} = -[M(\mathbf{x})]^{-1}\left(\partial M/\partial \mathbf{x}_{(k)}\right)[M(\mathbf{x})]^{-1}$

$$\frac{\partial \Phi_i(\mathbf{x})}{\partial \mathbf{x}_{(d)}} = [M(\mathbf{x})]^{-1}w_i(\mathbf{x})p(\mathbf{q}_i) - p^T(\mathbf{x})[M(\mathbf{x})]^{-1}\frac{\partial M(\mathbf{x})}{\partial \mathbf{x}_{(d)}}[M(\mathbf{x})]^{-1}w_i(\mathbf{x})p(\mathbf{q}_i) + p^T(\mathbf{x})[M(\mathbf{x})]^{-1}\frac{\partial w_i(\mathbf{x})}{\partial \mathbf{x}_{(d)}}p(\mathbf{q}_i)$$

From the definition of $M(\mathbf{x})$

$$\frac{\partial M(\mathbf{x})}{\partial \mathbf{x}_{(d)}} = \sum_{i=1}^{K} p(\mathbf{q}_i)p^T(\mathbf{q}_i)\frac{\partial w_i(\mathbf{x})}{\partial \mathbf{x}_{(d)}}$$

Where,

$$\frac{\partial w_i(\mathbf{x})}{\partial \mathbf{x}_{(d)}} = \begin{cases} \dfrac{\left(-3 \cdot (1 - ||\mathbf{x} - \mathbf{q}_i||_2/r_i)^2\right)}{(r_i \cdot ||\mathbf{x} - \mathbf{q}_i||_2)} \cdot (\mathbf{x} - \mathbf{q}_i) & \text{if } ||\mathbf{x} - \mathbf{q}_i||_2^2 \leq r_i \\ 0, & \text{otherwise} \end{cases}$$

Therefore, $\forall \mathbf{q}_i \ \ s.t. ||\mathbf{x} - \mathbf{q}_i||_2^2 \leq r_i$

$$\frac{\partial \Phi_i(\mathbf{x})}{\partial \mathbf{x}_{(d)}} = [M(\mathbf{x})]^{-1}w_i(\mathbf{x})p(\mathbf{q}_i)$$

$$-p^T(\mathbf{x})[M(\mathbf{x})]^{-1}\sum_{i=1}^{K}p(\mathbf{q}_i)p^T(\mathbf{q}_i)\frac{\left(-3 \cdot (1 - ||\mathbf{x} - \mathbf{q}_i||_2/r_i)^2\right)}{(r_i \cdot ||\mathbf{x} - \mathbf{q}_i||_2)} \cdot (\mathbf{x} - \mathbf{q}_i)(\mathbf{x})p(\mathbf{q}_i)$$

$$+p^T(\mathbf{x})[M(\mathbf{x})]^{-1}\frac{\left(-3 \cdot (1 - ||\mathbf{x} - \mathbf{q}_i||_2/r_i)^2\right)}{(r_i \cdot ||\mathbf{x} - \mathbf{q}_i||_2)}(\mathbf{x} - \mathbf{q}_i)p(\mathbf{q}_i)$$

$$\tag{1}$$

From Equation 1, we see that $\mathbb{J}$ is independent of the deformation parameter $u_i$. Since we know the node positions $\mathbf{q}_i$, in scenarios where evaluation points $\mathbf{x}$ are known, $\mathbb{J}$ can be pre-computed.

## 1.3 Architecture

A detailed overview of our architecture is show in Figure 1. We apply position encoding to our input coordinates following Tanick *et al.* [1]. Input coordinates are embedded to the surface of a 128 dimensional hypersphere.

## 1.4 Hyper-Parameters

In this section, we provide details on hyper-parameters, choice of template and nodes corresponding to each experiment in our main paper.

### 1.4.1 Shape Correspondence and Registration

We discuss the hyper-parameters used in Experiments 5.1 and 5.2 of the main paper respectively. Since our template is a mesh in these two experiments, we leverage the connectivity while sampling nodes in order to eliminate a node from influencing two vertices that belong to different semantic

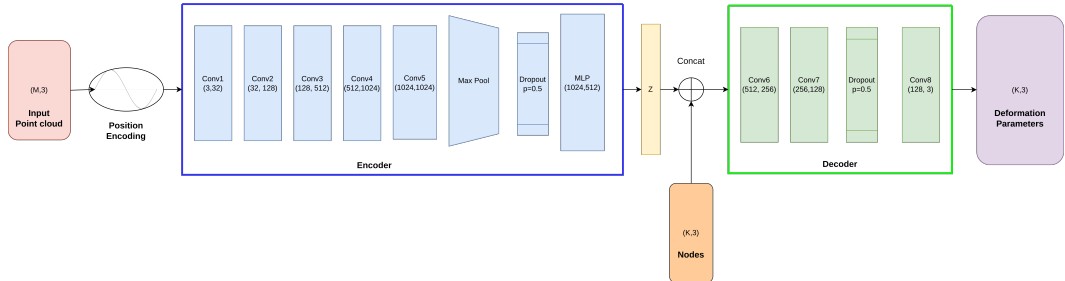

Figure 1: A detailed summary of our architecture

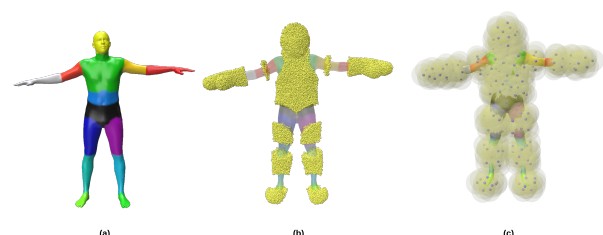

Figure 2: Different stages of node sampling strategy. (a) First, we segment the template using [2]. (b) Second, we sample points close to the surface of the template mesh called candidates. (c) Finally, we sub-sample from candidates until all vertices have 4 non-planar nodes in its vicinity. Nodes are shown as blue points with its region of influence in transparent yellow.

regions as shown in Figure 2. Hyper-parameters used during training and inference are given in Table 1

| Mode | Training | | | | | | Inference | | |
|---|---|---|---|---|---|---|---|---|---|
| Variable | K | $\lambda_1$ | $\lambda_2$ | $\lambda_3$ | $\lambda_4$ | $r_i$ | $\Lambda_1$ | $\Lambda_2$ | $\Lambda_3$ |
| Value | 300 | 1 | 5e-3 | 1e-2 | 5e-3 | 2e-1 | 1 | 1e-4 | 1e-3 |

Table 1: Hyper-parameters used for our non-rigid shape correspondence and registration experiments

### 1.4.2 Unsupervised Segmentation Transfer

The template point cloud and corresponding sampled nodes are shown in Figure 3. Since the shape collection exhibit significant structural difference, we slightly relax the first-order regularization. We perform test-time refinement for 200 steps. Our hyper-parameters are summarized in Table 2.

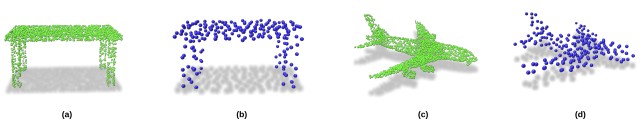

Figure 3: Template point clouds corresponding to Table and Plane class used in Experiment 5.3 from the main paper. (a) and (c) denotes the template while (b) and (d) denote sampled nodes.

| Mode | Training | | | | | | Inference | | |
|---|---|---|---|---|---|---|---|---|---|
| Variable | K | $\lambda_1$ | $\lambda_2$ | $\lambda_3$ | $\lambda_4$ | $r_i$ | $\Lambda_1$ | $\Lambda_2$ | $\Lambda_3$ |
| Value | 300 | 1 | 1e-4 | 1e-3 | 1e-4 | 2e-1 | 1 | 1e-4 | 1e-3 |

Table 2: Hyper-parameters used in unsupervised segmentation transfer experiment.

### 1.4.3 Shape Interpolation

We do not perform test-time refinement for this experiment. Table 3 summarizes the training hyper-parameters. We use the same nodes as in shape correspondence experiment.

| Mode | Training | | | | | |
|---|---|---|---|---|---|---|
| Variable | K | $\lambda_1$ | $\lambda_2$ | $\lambda_3$ | $\lambda_4$ | $r_i$ |
| Value | 300 | 1 | 1e-4 | 1e-2 | 1 | 2e-1 |

Table 3: Hyper-parameters used in shape interpolation experiment.

## 2  Comparison with Alternatives

We perform quantitative comparison between our approach and plausible alternatives in terms of accuracy and computational efficiency. We compare with two main alternative representations. First, deformation at each point in the template shape is predicted using a point-wise MLP, abbreviated as PW-MLP. Second, we compare with an interpolation variant, where we replace the mesh-free approximation with Radial Basis Function (RBF) interpolation. We provide a brief overview of this interpolation variant below. In order to have a controlled setting, all our experiments are performed on a machine with Nvidia Ampere A100 GPUs and AMD 7302 3Ghz CPU over same training, batch-size and number of points used as input to the encoder.

**RBF Interpolation**  Instead of *approximating* the deformation field across the surface using mesh-free functions, we use RBF interpolation to *interpolate* the deformation field based on observations at nodes. Our main pipeline (as shown in Figure.1 of main paper) remains the same, barring the fact that, here, we interpolate the deformation field. Let $U$ be the predicted deformation parameters at nodes $\mathcal{Q}$, $\varphi(\cdot)$ be the RBF kernel, $\Phi$ to be its matrix representation, interpolant function $f(\cdot)$ at a point $\mathbf{x} \in \mathbb{R}^3$ is given as:

$$f(x) = \sum_{i=1}^{K} \Phi^{-1} U \varphi(\mathbf{x}, \mathbf{q}_i). \tag{2}$$

Here $\mathbf{q}_i \in \mathcal{Q}$ and K denotes the total number of nodes. The kernel function $\varphi(\cdot)$ [3] and its matrix representation $\Phi$ are defined as:

$$\Phi_{mn} := \varphi(\mathbf{q}_m, \mathbf{q}_n) = \sqrt{C + \epsilon_0 ||\mathbf{q}_m - \mathbf{q}_n||^2}, \tag{3}$$

where, $m$ and $n$ denote the $m^{th}$ row and $n^{th}$ column of $\Phi$ respectively, $\mathbf{q}_n, \mathbf{q}_n \in \mathcal{Q}$. Please note that $\Phi$ is positive definite by its construction. $C = 1$ and $\epsilon_0 = 50$ are taken to be constants. Similar to the case of mesh-free approximation, it is easy to see that evaluation of Jacobian is independent of values of deformation field itself:

$$\mathbb{J} = \nabla_{x,y,z} f(\mathbf{x}) = \Phi^{-1} U \left( \sum_{i=1}^{K} \left[ \frac{\partial \varphi(\mathbf{x}, \mathbf{q}_i)}{\partial x}, \frac{\partial \varphi(\mathbf{x}, \mathbf{q}_i)}{\partial y}, \frac{\partial \varphi(\mathbf{x}, \mathbf{q}_i)}{\partial z} \right] \right) \tag{4}$$

**Interpolation vs Approximation**  Although interpolation is close to our proposed representation in terms of effective reduction, there are three key differences between both representations. First, the weighting function expressed through $\varphi(\cdot)$ has an *infinite support* while our approximation has compact support. Second, our representation guarantees the approximation function of $n^{th}$ order consistency depending on the polynomial basis. Third, in the case of interpolation, $f(\mathbf{q}_i) = u_i | u_i \in U$, whereas while approximating, $u(\mathbf{q}_i) \neq u_i | u_i \in U$. This distinction is important from the meaning it endows our network $\mathcal{F}(\cdot)$. In the interpolation case, it amounts to predicting the *deformation field* whereas in the approximation case, $\mathcal{F}(\cdot)$ predicts *deformation parameters*.

| Constraint | None | | | $\mathcal{L}_{vol} + \mathcal{L}_{arap}$ | | | $\mathcal{L}_{vol} + \mathcal{L}_{arap} + \mathcal{L}_Z$ | | |
|---|---|---|---|---|---|---|---|---|---|
| Method | PW-MLP | RBF-Inp | Ours | PW-MLP | RBF-Inp | Ours | PW-MLP | RBF-Inp | Ours |
| Time(iter/ms) | 32.0 | 8.0 | 7.8 | 250.4 | 8.2 | 8.0 | 4098.3 | 12.6 | **12.1** |
| SCAPE (PC+N) | 13.7 | 9.9 | 9.8 | 12.6 | 9.1 | 7.0 | 14.8 | 9.3 | **6.6** |
| SHREC19 | 7.4 | 8.5 | 7.7 | 7.1 | 8.0 | 5.2 | 9.1 | 8.1 | **4.8** |

Table 4: Quantitative comparison of efficiency and correspondence accuracy between possible alternative representations and our approach. Imposing latent-space regularization results in prohibitive computation effort using a standard PW-MLP representation of deformation fields.

**Discussion** We summarize the quantitative and qualitative results in Table 4. Our approach shows nearly $325\times$ improvement in speed when applying first-order regularization compared to the PW-MLP. While both reduced variants, namely, interpolation and approximation show comparable timings, there is, however, a significant performance difference between them in terms of correspondence accuracy. We attribute this observation mainly to the compactness of our approximation. Without such compactness, deformation fields corresponding to different semantic regions may influence one another. For instance, the deformation field corresponding to left arm of a human can affect the way the right arm moves or the "pull-effect". This particular example is characterized in Figure 4, where *interpolating* the deformation field fails to capture the articulation. In addition, the "pull-effect" also shortens the length of the left-arm in comparison to the right.

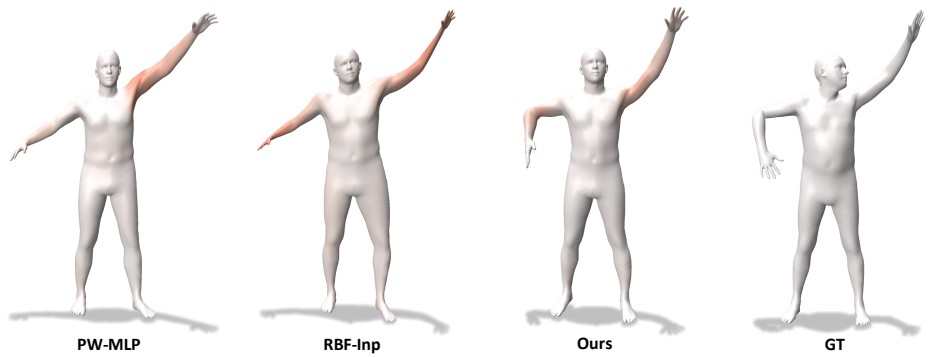

Figure 4: Qualitative comparison between Point-wise MLP(PW-MLP), RBF interpolation (RBF-Inp) and our approach (Ours). Meshes are color-coded with area distortion. Increasing shades of red signifies larger distortion.

## 3  Ablation Study

In this section, we perform an in-depth ablation study to analyze the effect of first-order regularization during training and inference. Then, we ablate our choice of number of nodes, its influence, different positioning strategy and pose of template used. Training regularization (Tr-Regularization) and Test-time regularization (Te-Regularization) refers to Equation 6 and Equation 11 from the main paper respectively. We re-train all methods on the same 1000 SURREAL shapes [4] mentioned in the main paper while evaluating them on SHREC'19 [5] and SCAPE (PC+N) datasets for the non-rigid shape correspondence task.

| Dataset | Tr-Regularization | | | | Te-Regularization | | Ours |
|---|---|---|---|---|---|---|---|
| | None | $\mathcal{L}_{arap}$ | $\mathcal{L}_{vol}$ | $\mathcal{L}_{vol} + \mathcal{L}_{arap}$ | None | $\mathcal{L}_{CD}$ | All |
| SHREC'19 | 7.7 | 6.9 | 7.3 | 5.2 | 7.9 | 5.0 | **4.8** |
| SCAPE | 9.8 | 10.2 | 8.7 | 7.0 | 12.5 | 6.7 | **6.6** |

Table 5: Ablation study on regularization at training and test time. All training regularization are imposed when ablating test-time regularization and vice-versa.

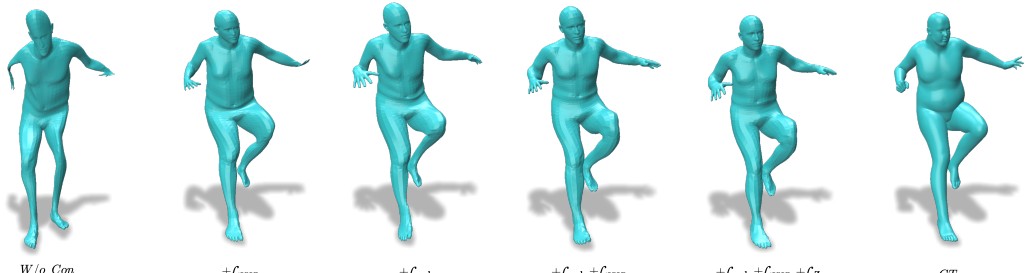

W/o Con  $+\mathcal{L}_{arap}$  $+\mathcal{L}_{vol}$  $+\mathcal{L}_{vol} + \mathcal{L}_{arap}$  $+\mathcal{L}_{vol} + \mathcal{L}_{arap} + \mathcal{L}_Z$  GT

Figure 5: Effect of different regularization applied to the deformation field. While enforcing $\mathcal{L}_{arap}$ reconstructs the shape, the lack of $\mathcal{L}_{vol}$ leads to "collapse" effect at hands and legs. Similarly using $\mathcal{L}_{vol}$ does not result in a distortion-free reconstruction. Finally, incorporating $\mathcal{L}_Z$ produces distortion-free deformation at joints (see right elbow).

## 3.1 Training Regularization

Our main observations on the efficacy of deformation field regularization is summarized in Table 5. Although the mapping between the template shape and target are highly non-isometric, yet incorporating first-order regularization show a notable improvement in accuracy.

This is because we do not restrict ourselves to exactly volume preserving deformations, but rather use our regularizers to penalize implausible deformations, that can incur significant volume distortion. We empirically observe such regularization helps in producing better results especially in the presence of limited training data. We also show an example of deformed templates corresponding to the losses we ablate in Figure 5. While all variants of our method recover the pose of the template, the reconstructions are plausible only when enforcing first-order regularization.

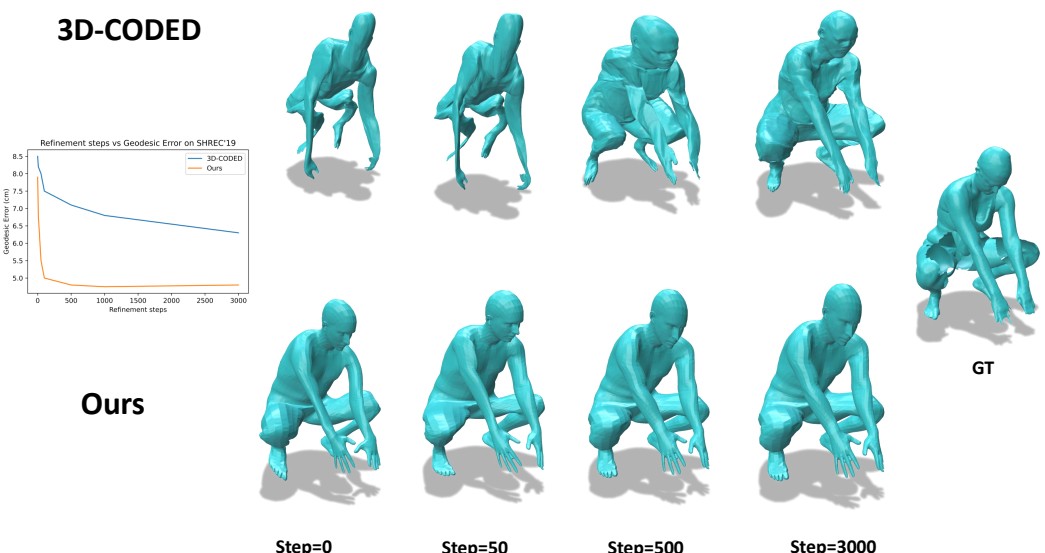

Step=0  Step=50  Step=500  Step=3000

Figure 6: Quantitative and qualitative illustration of test-time refinement. Our demonstrates a plausible reconstruction w/o refinement which baseline fails to accomplish. This results in requiring significantly less test-time refinement efforts as compared to 3D-CODED.

## 3.2 Test-time Regularization

We perform test-time refinement to enhance the reconstruction similar to 3D-CODED [6]. However, owing to our structured deformation space, our method provides a more plausible initialization,

thereby requiring significantly less refinement steps as shown in Figure 6. This is particularly beneficial in expediting the inference process.

| # Nodes | 100 | 300 | | 900 | | | 2700 | | |
|---------|-----|-----|-----|-----|-----|-----|------|-----|------|
| Radius | 0.5 | 0.2 | 0.5 | 0.1 | 0.2 | 0.5 | 0.07 | 0.2 | 0.5 |
| SHREC'19 | 6.8 | **4.8** | 5.4 | 5.4 | 6.4 | 6.5 | 6.2 | 6.5 | 6.5 |
| SCAPE-PC | 8.2 | **6.6** | 7.6 | 7.0 | 8.7 | 8.8 | 9.8 | 9.6 | 10.1 |

Table 6: Ablation study on number of nodes and radius. Radius is expressed as fraction of shape diameter. Errors on two benchmarks are reported in cm.

### 3.3 Optimal nodes and radius

The number of nodes and their radius are important parameters in our reduced representation. We desire a representation that is both compact and simultaneously can capture the local characteristics of deformation. Unfortunately, both of these criteria are difficult to satisfy simultaneously as it could potentially lead to a singular moment matrix (c.f Eqn 2, main draft). Therefore, we first make a choice on the compactness by letting the radius of each node be $\frac{1}{5}^{th}$ of the shape diameter. Then, from the set of candidates (Figure 2(b)) we sample nodes until the non-singularity condition for the moment matrix is satisfied. Since the choice of radius is a hyper-parameter, we perform an ablation study by varying the radius and the number of nodes as summarized in Table 6. We observe that by increasing the radius of each node and the number of nodes itself deters the performance. This is because larger radius impedes the *locality* of the deformation by influencing distant points. On the other hand, increasing the number of nodes forces the network to learn more fine-grained details thereby showing a deteriorated performance in the setting of limited training data. Alternatively, one could first choose a fixed set of nodes and then expand their radii until the non-singularity condition is met. However, this requires a prior-knowledge on where to place the nodes and that requires manual intervention. On the other hand, our selection process is fully automatic.

### 3.4 Positioning of nodes

Our motivation behind segmenting the template at the time of node initialization is to prevent a node from influencing the deformation field over sets of points on the template that are far from each other in a geodesic sense as this could lead to non-local deformation behavior. Since our template is from the SURREAL [4] dataset, we leveraged the SMPL segmentation in our main draft. In this section, we explore three alternative sampling strategies. First, we segment the template by performing K-means over the first 4 eigenvectors of the Laplace-Beltrami Operator of the template mesh as shown in Figure 7. This is a well-known segmentation technique in Computer Graphics introduced by Rustamov [7]. This technique is unsupervised in the sense it assumes no knowledge of mesh vertex ordering. Second, we reject nodes which can influence a pair of points between which geodesic distance is larger than 20% of shape diameter. Our final baseline is a simple Farthest Point Sampling (FPS) over the dense point cloud sampled over template mesh without any rejection. We compare different sampling techniques in Table 7 over two template meshes, namely in A-pose and T-pose respectively. We observe that our proposed sampling strategy is more effective for a template in A-pose while showing marginal improvement over a straightforward sampling for a template in T-pose. This is because the likelihood of a node to influence geodesically farther (or semantically different) points is significantly higher in A-pose than in T-pose. We illustrate an example in Figure 7 where a node from uniform sampling is shown to influence the deformation field at both arms and torso. We note, however, that the approach based on unsupervised segmentation, performs similarly to our strategy and does not require any prior semantic information.

| Pose of Template | T-Pose | | | | A-Pose | | | |
|------------------|--------|--------|------------------|-----------------|--------|--------|------------------|-----------------|
| Sampling Strategy | Uniform | Geodesic | Segmentation SMPL | Segmentation [7] | Uniform | Geodesic | Segmentation SMPL | Segmentation [7] |
| SHREC'19 | 5.1 | 5.0 | **4.8** | 4.8 | 5.9 | 5.2 | **5.1** | **5.1** |
| SCAPE | 6.9 | 6.9 | **6.6** | 6.6 | 9.5 | 7.4 | 7.4 | **7.3** |

Table 7: Comparison of different sampling strategy for initializing nodes across two template poses. All reported errors are in cm.

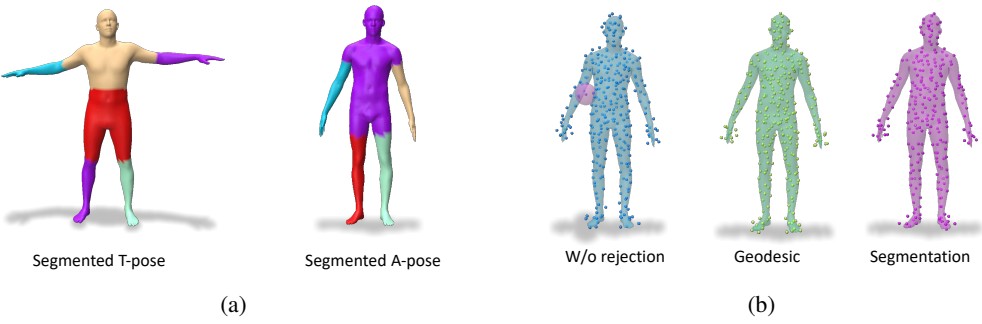

| Segmented T-pose | Segmented A-pose | W/o rejection | Geodesic | Segmentation |
| --- | --- | --- | --- | --- |
| (a) | | (b) | | |

Figure 7: (a) denotes pose invariant segmentation [7] done over T-pose and A-pose template respectively. (b) Denotes various ways of sampling nodes over A-pose template shape. Uniform sampling leads to nodes influencing points that are far in geodesic sense (highlighted in purple). This artifact is avoided by using geodesic distance on template mesh or segmentation information. Underlying surface is rendered for visualization purpose only.

## 3.5 Choice of Template

We analyze our choice of template by comparing with three alternatives which have different pose and style as shown in Figure 8. We compare shape-correspondence accuracy on the SHREC'19 [5] benchmark. Template in A-pose and T-pose showed comparable performance while it mildly deteriorated when using an I pose template.

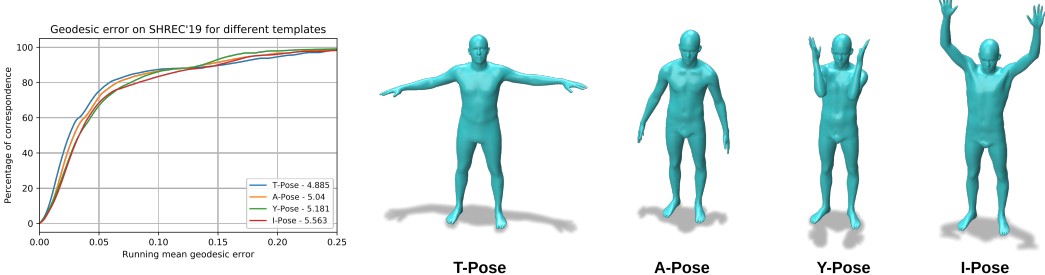

Figure 8: Quantitative correspondence result on SHREC'19 [5] (left) using different templates (right).

# 4 Effect of supervision

## 4.1 Optimal training shapes

We simultaneously decrease and increase the amount of training data to analyze the data-dependence of various supervised deformation methods. To that end, we train our method and baselines on 250, 500, 2000 training shapes sampled at random from SURREAL dataset [4]. We re-train the deformation baselines TransMatch [8] and 3D-CODED [6] on the same dataset with appropriate parameters for a fair comparison. Figure 9 summarizes our comparison. Our approach shows significant improvement over 3D-CODED [6] when trained on 250 and 500 shapes across both SCAPE (PC+N) and SHREC'19 benchmarks respectively. TransMatch [8] on the other hand fails to achieve reasonable correspondence due to the strong reliance of attention mechanism on large collection of training data.

## 4.2 Optimal corresponding points

We analyze the need for supervision by comparing with 3D-CODED by varying the number of points used for supervision (c.f Equation 7 main paper). We use 50, 100 and 1000 points for supervision and compare on SCAPE (PC+N) and SHREC'19. For a fair comparison, we re-train 3D-CODED on

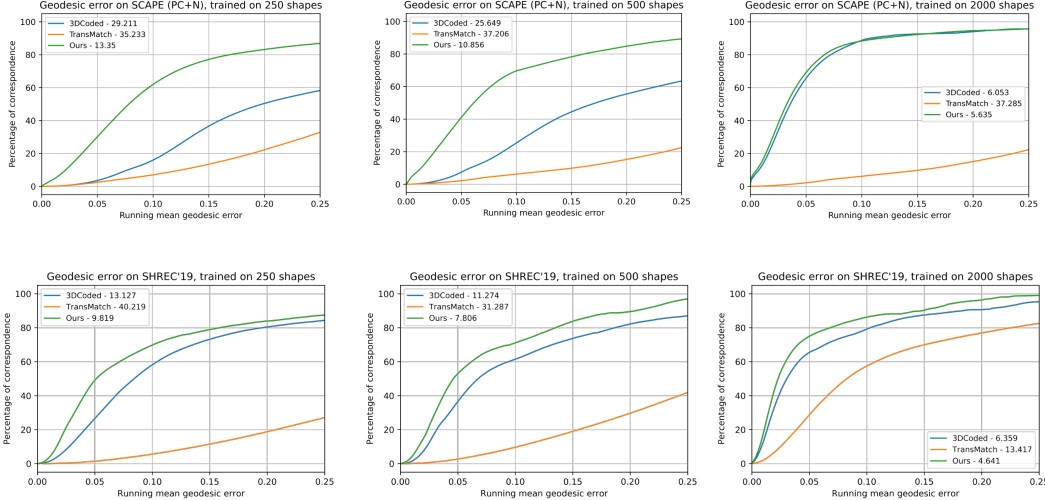

Figure 9: Quantitative correspondence accuracy with varying number of shapes in the training set. Our approach shows a significant improvement over the baseline particularly when there is paucity of data.

same training shape as our method. Results are summarized in Figure 10. It is remarkable to note that our method shows improvement over the baseline with one-tenth of supervision.

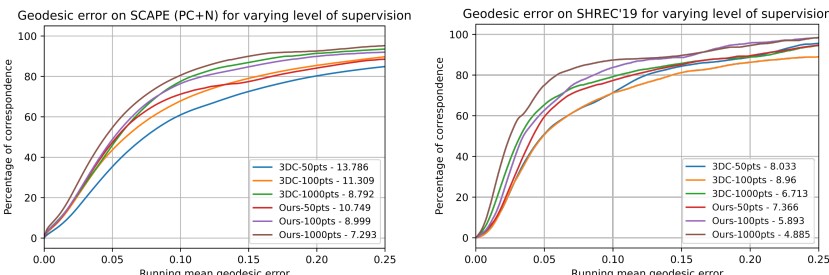

Figure 10: Quantitative comparison between our approach and 3D-Coded [6] with varying level of supervision used

# 5 Unsupervised Implicit Shape Correspondence

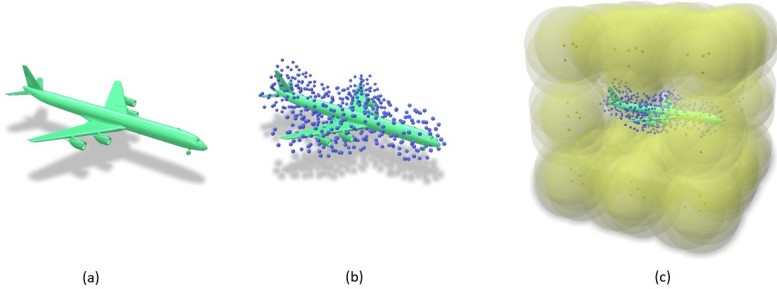

Figure 11: Node sampling strategy for modelling deformation between implicit fields. (a) Given a template mesh, (b) we sample nodes near the surface. (c) Then, we define "auxiliary" nodes along the vertices and edge of the unit cube. Auxiliary nodes are shown in red and its spatial influence in yellow (transparent).

Complimentary to discussions in Section 5.3 of our main paper, we demonstrate that our proposed representation is data-efficient in establishing correspondence between learnt implicit surfaces of real-world objects from ShapeNet [9] dataset. For this task, we adapt the existing implicit shape correspondence work DIF-Net [10] and replace their point-wise "Deform-Net" with our reduced representation. This replacement however is not straightforward as a *volumetric* deformation requires the deformation field to be continuous and defined in $\mathbb{R}^3$ while our compact representation restricts to a sub-region $\Omega \subset \mathbb{R}^3$. To overcome this, we scale all shapes to fit a unit cube and place *auxiliary nodes* at each vertex of the cube. These auxiliary nodes have a larger radius, thereby covering the entire region as shown in the Figure 11. Please recall that since the deformation field at a point varies inversely by its distance from neighbouring nodes, the influence of auxiliary nodes are minimal near the surface of the shape. As a result, auxiliary nodes act as a regularizer for points that are far from the surface.

We consider three categories namely chair, table and plane. Since there is no dense ground-truth correspondence annotation between them, we show qualitative interpolation results. For a fair comparison, we train both DIF-Net and our method over same shapes, consisting of 500 random samples from each category. We summarize our qualitative results in Figure 12. Our method produces *plausible* interpolation sequences owing to the latent constraint $\mathcal{L}_Z$ that regularizes the deformation field corresponding to intermediate shapes. In addition, our deformation field is smooth as shown by the color-map.

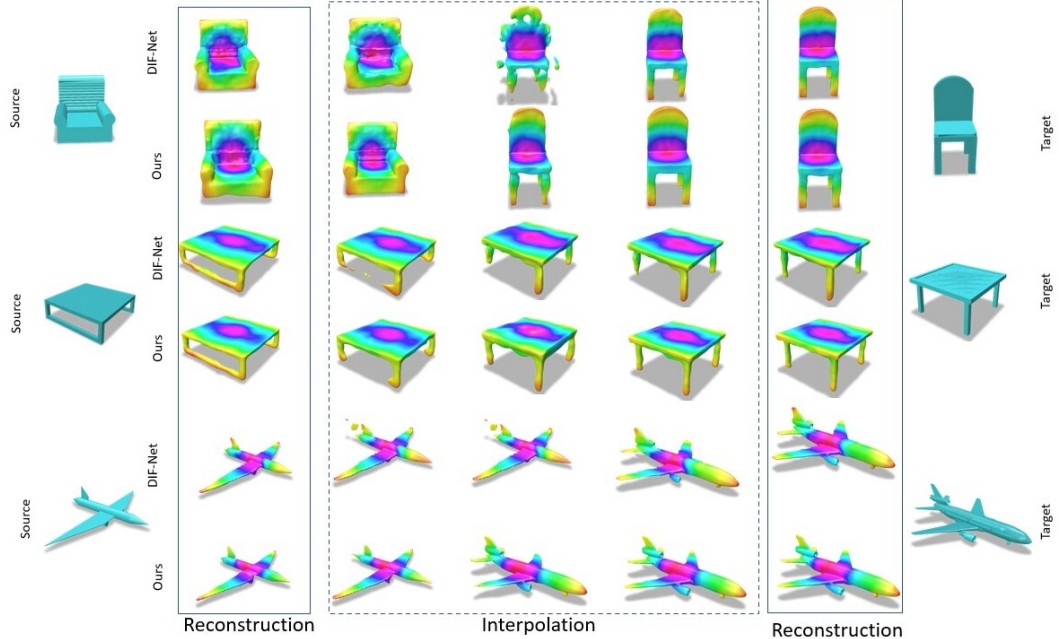

Figure 12: Qualitative comparison for implicit shape correspondence across three object categories. Individual columns in interpolation corresponds to samples at same time-steps across rows. Our well-structured latent space helps to avoid implausible reconstructions.

# 6 Real-world Data

Finally, we demonstrate the generalization of our approach in solving correspondence between real-world data over two datasets, namely, point cloud from RGB-D scans of humans and meshes of hearts.

## 6.1 CMU Panoptic Dataset

We show qualitative results of texture transfer between point clouds obtained from *Kinect RGB-D sensor*, from the CMU Panoptic dataset [11]. Our results are shown in Figure 13. Last row shows a

particularly interesting example of texture transfer between different subjects. Our approach is the only method that provides reasonable correspondence.

## 6.2 Human Heart Meshes

In order to demonstrate the applicability of our novel deformation-field representation beyond articulated non-rigid shapes, we consider the publicly available virtual cohort of four-chamber heart meshes dataset [12]. This dataset is generated from twenty-four heart failure (HF) patients, starting from CT scan images of the heart. These images are then segmented to distinguish the four chambers and a tetrahedral mesh is constructed from the resulting segments. In our experiments, we use the outer boundary surface of the mesh after Quadratic Edge Collapse Decimation [13]. Qualitative results are shown in Figure 14. We compare our approach with ZoomOut [14], an axiomatic shape correspondence technique and our closest deformation-based baseline, 3D-CODED [15]. Our approach extracts a more consistent correspondence in comparison to the two baselines. We attribute this to our regularization that facilitates smoothness and learning over a small-scale dataset.

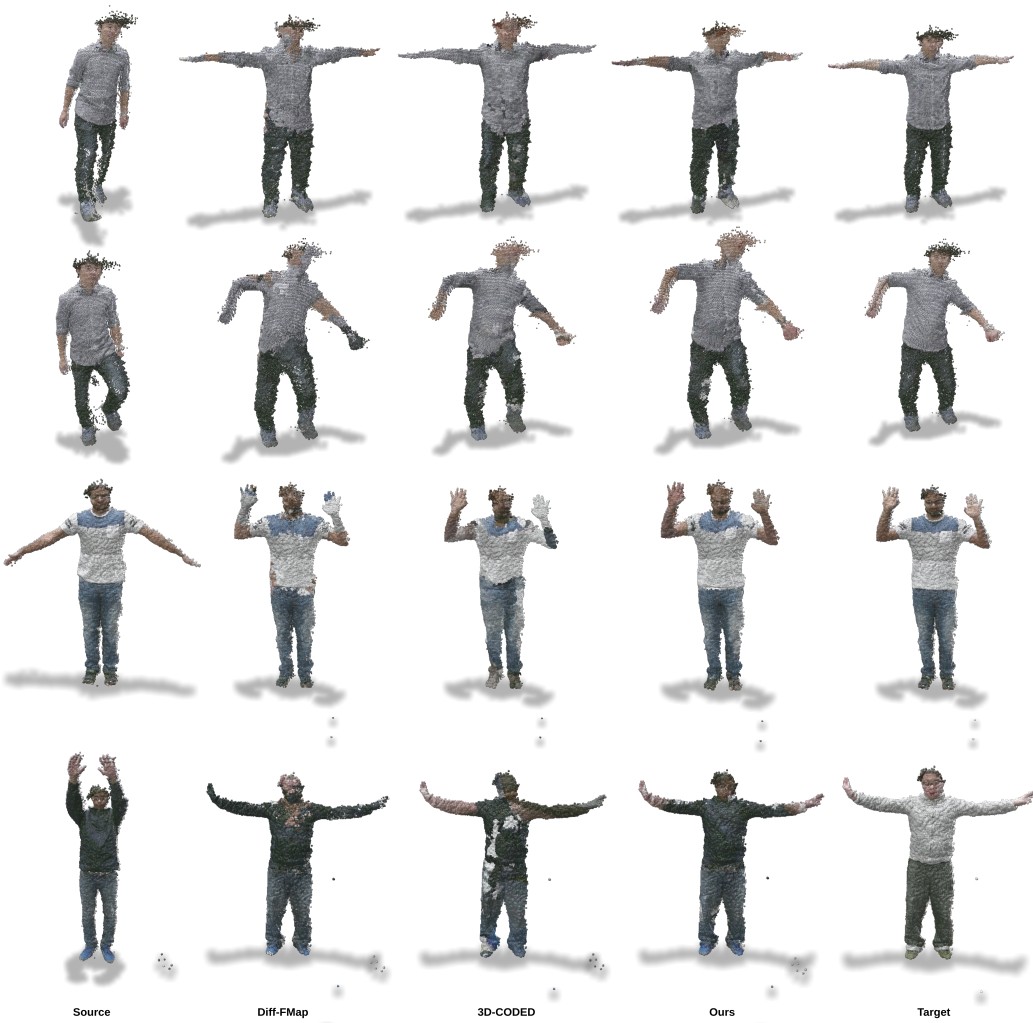

Figure 13: Qualitative comparison on real-world data from the CMU Panoptic dataset with noise and outlier points. First three rows show texture transfer between same subjects while the last row is an example of inter-subject transfer.

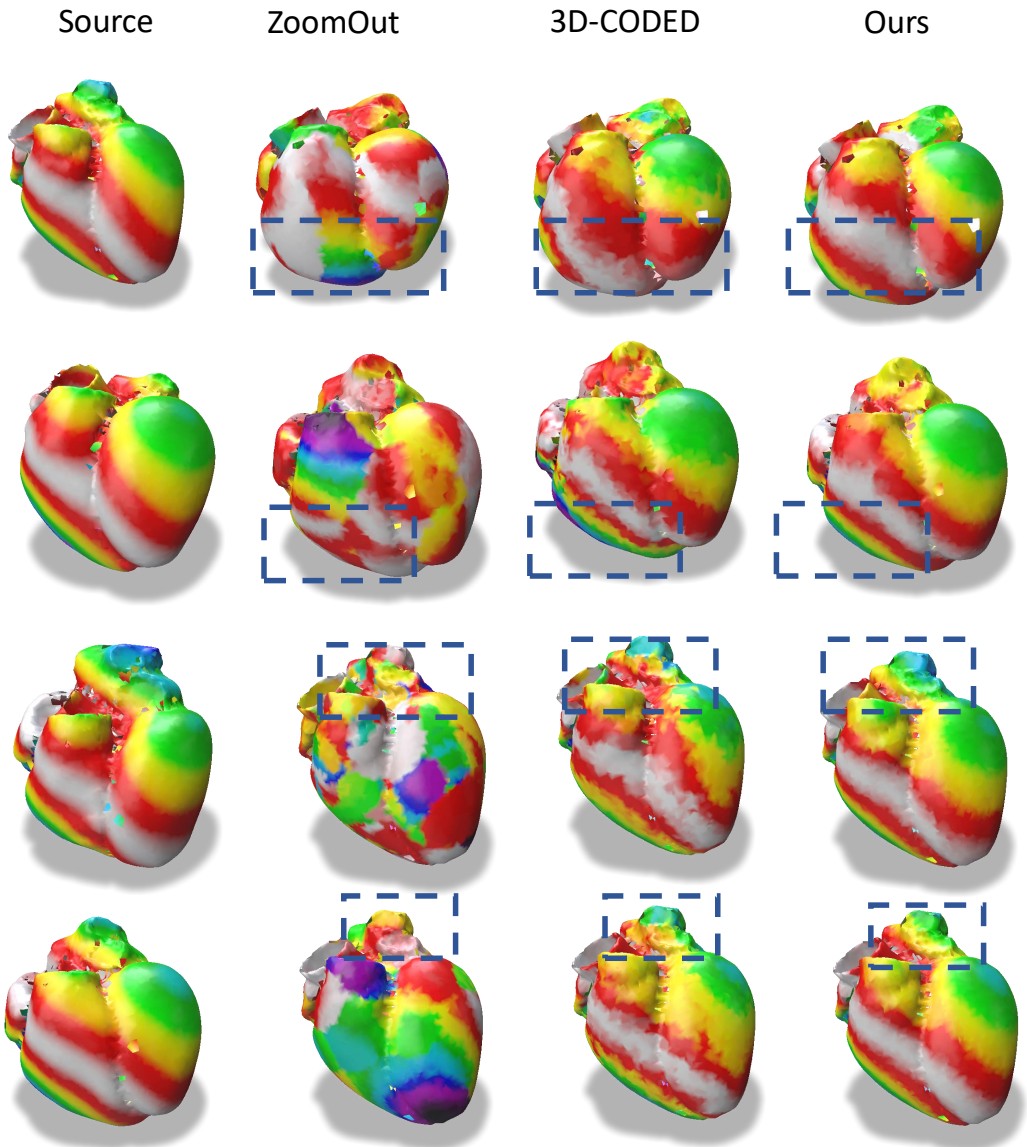

Figure 14: Qualitative comparison of colour-transfer between human hearts of different subjects obtained from [12]. Dotted box highlights the efficacy of our approach in computing smooth and consistent map.