# OpenReview forum: "Reduced Representation of Deformation Fields for Effective Non-rigid Shape Matching"
_NeurIPS.cc/2022/Conference — NeurIPS 2022 Accept_

### Official Review · Reviewer_86v4 · 2022-07-07

**Rating:** 4
**Confidence:** 3
**Soundness:** 2 fair
**Presentation:** 3 good
**Contribution:** 2 fair

**Summary:**

The paper computes the correspondences between two non-rigid 3d objects with point cloud or mesh representations. From given 3D inputs with only limited supervision, the proposed method effectively understands the correspondences and matches suitable parts between two different 3D objects. The authors proposed how to represent a parametrized compact deformation in the problem of 3D matching. In the experimental section, the authors have shown the effectiveness of their proposed method in terms of non-rigid shape matching, registration, unsupervised part segmentation, and interpolation while comparing to each baseline method. The paper is also positive to publicly share their codes and models proposed in the paper.


**Questions:**

- In the end, does the matching be done by 3D Point Cloud instead of mesh? In case of mesh input, will the conversion must be processed in order to translate 3D Point Cloud?

- Why does the compact representation reduce the number of training samples? It would be better to have a discussion in the experimental section, even if it is just a citation of related studies and their discussion.


**Limitations:**

- I can understand the paper limitation due to 9-page restriction, however, the discussion part is not sufficient in order to judge the effectiveness in the proposed method. At the current revision, the authors list the fact of results from performance rates and qualitative results in figures. More detailed discussion should be required such as "why is the proposed method effective in the category 'twisted'", and "why can the proposed method train with a limited labeled 3D data". I believe that a more detailed discussion improves the paper quality.


**Strengths And Weaknesses:**

[Strengths]
S1) The MLP-based representation approximates fixed-length parameters. The compact 3D representation matches a high-level matching between two different objects, despite the small number of training samples are worthy of evaluation. By using the proposed method, the authors have achieved good performance rates in comparison with conventional methods and is expected to become a good baseline in this field.

S2) Experimental comparisons have been made with the respective baseline methods in diverse tasks (non-rigid shape matching, registration, unsupervised part segmentation, and interpolation) to demonstrate their superiority. These comparisons show the effectiveness of the proposed method in wide-range of tasks.

S3) The code distribution in itself has a positive impact to the 3D matching community. In terms of reproducing a 3D matching method that is compact yet achieves high performance, it is easier for readers to try the proposed method. Additionally, the code would make the paper more understandable.

[Weaknesses]
W1) While the proposed paper has very good properties - a compact feature representation, small amount of training samples, and higher matching performance - there exists weaknesses that the detailed parameters are not well demonstrated through experimental section.

W2) Although the number of training samples are suppressed which is a good aspect in the paper, the lack of a relationship between the #training-sample and their performance is insufficient. There is no guarantee as to how much training sample is required in order to reach a sufficient level of 3D matching performance. The authors can appeal to the research community if the proposed method enables to train a well-deserved 3D matcher from extremely limited number of training samples.

W3) Eq. (6) shows the objective functions, however, the mathematical equation is not validated in terms of their combination and contribution to the training at each loss function. It would be better if the loss functions of importance are clarified.

---

> ### Author Response · Authors · 2022-08-02
> **Response to reviewer 86v4**
>
> 1. > **The detailed parameters are not well demonstrated through experimental section.**
>
> We have updated our ablation studies (Sec 3.3, Sec 3.4 of Suppl) by elaborating the impact of different parameters with respect to our representation such as 1) Number of Nodes, 2) The radius of impact and 3) How to properly sample nodes. If anything is still ambiguous, we will be happy to clarify further.
>
> 2. > **the lack of a relationship between the #training-sample and their performance.**
>
> We kindly invite you to refer to Sec 4 of Suppl, where we vary the number of training shapes and the number of supervision points used at training time. We compare our performance with two relevant baselines - 1) 3D-CODED which uses a standard pointwise MLP to predict deformation and 2) TransMatch which uses a Transformer architecture to predict the deformation. The input (point cloud) and output (deformed shape) of all three methods are the same.
>
> 3. > **There is no guarantee as to how much training sample is required in order to reach a sufficient level of 3D matching performance.**
>
> While a theoretical guarantee would be ideal, to the best of our knowledge, we know of no such measure in the domain of 3D Shape Matching/Registration. Therefore, we use the same empirical analysis performed in prior works such as [1,2] by providing a quantitative insight by varying the number of training samples.
>
> 4. > **It would be better if the loss functions of importance are clarified.**
>
> We kindly invite you to refer to Sec 3.1, Sec 3.2 of Suppl, where an extensive ablation for the importance of each term of the loss function at both training and inference time has been provided (also see Tab 5 and Fig 5 of the Suppl). The importance of each term is provided in Section 1.4 of Suppl, where, for each experiment, we provide all the hyper-parameters associated with it. If there is something which we have not addressed, please clarify further and we will be happy to elaborate.
>
> 5. > **In the end, does the matching be done by 3D Point Cloud instead of mesh?**
>
> Our method can handle both point clouds and meshes or even find a correspondence between a point cloud and a mesh. This is because our learning pipeline (encoder in particular) is invariant to the connectivity and permutation[3] of the points in the target shape.
>
> 6. > **In case of mesh input, will the conversion must be processed in order to translate 3D Point Cloud?**
>
> The only preprocessing we do over the geometry is scaling to fit within a unit sphere as provided in the Section 1.1 of the Supplementary. This is straightforward with O(N) complexity.
>
> 7. > **Why is the proposed method effective in the category 'twisted'**
>
> We used the “twisted” pose as an illustration of the difficulty of the problem, and can be seen as a stress-test, due to the complexity of articulation. While there might exist other complicated poses, the particular characteristic of this pose is that a wrong deformation can lead to large geodesic error. For example, the hand can be mapped to the trunk by the “cheese-pull” effect (see [4], Figure 11 for an illustration of what this implies). We hypothesize that our approach, which learns a “global” sense of the articulation, does not suffer from such artifacts since the fine (local) details are computed in closed form. This is in contrast to other baselines reported in Fig 2, main draft. We will elaborate this reasoning in the discussion section in our final version.
>
> 8.> **Why can the proposed method train with a limited labeled 3D data?**
>
> We have addressed this in our general comment (above). We will be happy to clarify further if there is any ambiguity in our explanation.
>
>
> ### References
>
> [1] Donati et.al. "Deep Geometric Functional Maps: Robust Feature Learning for Shape Correspondence", CVPR 2020
>
> [2] Eisenberger el.al. "Deep Shells: Unsupervised Shape Correspondence with Optimal Transport", NeurIPS 2020
>
> [3] Qi et.al PointNet: "Deep Learning on Point Sets for 3D Classification and Segmentation", CVPR 2017
>
> [4] Eisenberger et.al. "Smooth Shells: Multi-Scale Shape Registration with Functional Maps", CVPR 2020

---

### Official Review · Reviewer_DTPf · 2022-07-11

**Rating:** 6
**Confidence:** 4
**Soundness:** 4 excellent
**Presentation:** 3 good
**Contribution:** 4 excellent

**Summary:**

This work addresses the issue of attaining correspondence between non-rigid objects.

Conventionally, methods using intrinsic representation require accurate mesh for each object, whereas extrinsic methods require various annotated data for training to achieve accurate correspondences. The authors propose to take an intermediate approach and use ''nodes'', or fixed control points that determine the deformation of the object.

To achieve this goal, they define the influence of each nodes using a shape function based on moving least squares, an established deformation model, and propose to learn the corresponding deformation parameters that determine the shape. One of the shapes is selected as a template shape, and the deformation parameters required from the template to each target is learnt.

To constrain the learnt deformation, the paper proposes to add various constraints using the Jacobian of the deformation field, which can be precomputed for the template shape, making the calculation significantly efficient. Furthermore, they add a constraint on the latent space so that intermediate shapes between the template and the target can be generated by traversing a continuous path in the resulting latent space.

**Questions:**

What kind of poses would be difficult to be used as the template?

I am curious as to why the data-heavy baselines such as 3D-CODED cannot achieve high correspondence accuracy compared to the proposal. The authors mention deformation requiring less parameters, but the deformation model used in the method is rather conventional and presumably less flexible compared to the learnt one. What was the key difference? Is the lack of constraints in the prior methods the main source of weakness?

In Section 4.2, the authors mention that correspondence between unknown shapes $(\mathcal{X}, \mathcal{Y})$ is found by extracting mappings $(\mathcal{D}_\mathcal{X},\mathcal{D}_\mathcal{Y})$ and then finding the closest neighbors. However, if I understood correctly, doesn't Dx and Dy represent deformation required for template $\mathcal{T}$ to deform to $\mathcal{X}$ and  $\mathcal{Y}$? If so, the end poses are different and nearest point would unlikely be true correspondences. I think the inverse transformation is required to establish correspondence. I would like some clarification regarding the test phase. Or, does this mean that $\mathcal{D}_\mathcal{X}$and $\mathcal{D}_\mathcal{Y}$ are templates transformed to \mathcal{X} and \mathcal{Y}, and closest points corresponding to each point on the deformed templates in $\mathcal{D}_\mathcal{X}$and $\mathcal{D}_\mathcal{Y}$ are considered correspondences? This would make sense but is not very obvious from the text.

In many cases involving human figures, the head does not seem to be turning. Is there any particular reason for this phenomenon?

What would happen if the nodes were more evenly distributed, without removing any from near the segment boundaries? Would the experimental results be worse?

**Limitations:**

The authors do disucss the limitation of the proposal. One is the bijectivity of the correspondence, and the other deals with the fixation of the nodes.

I would also like to see what would happen for shapes whose parts are difficult to determine (e.g. human organs, etc.), as the term ''non-rigid'' includes such objects as well. In other words, the title seems to suggest broader application, whereas the paper rather mainly targets articulated shapes.

**Strengths And Weaknesses:**

### Strengths

I was very intrigued by how the authors utilized well-established algorithms and models in conventional non-rigid shape analysis works, such as moving least squares and constraints using the Jacobian of deformation fields, and focus the learning on the corresponding deformation parameters, and not the point coordinates themselves. This enables the proposed method to effectively recover deformation parameters around the fixed nodes, which can be diffused to the surrounding regions to estimate the deformation field around the entire shape. Various experiments show that the resulting correspondence is highly accurate, indicating that the deformations were properly conducted.

The overall accuracy, especially with respect to achieving proper correspondence among different poses is very impressive. I presumed that data-driven extrinsic methods would perform rather well, however, according to the manuscript, the proposal is able to achieve highly accurate correspondence while using less data for training. This approach seems useful in practice, as it does not require accurate meshes.

The continuous constraint on the latent space is also interesting, as it enabled smooth transition between each of the source pose and the template. The intermediate poses in Fig. 5 demonstrate the effectiveness of such constraint. The authors also thoroughly demonstrated the role of each constraint in the supplementary material, which further supported the necessity of each element.

### Weaknesses

The node sampling strategy in Section 4.4 seems to imply that the deformation is mainly articulation-based, rather than a fully non-rigid one. I was actually curious about why this segmentation-based sampling is required, as I presumed that the deformation function based on moving least squares would nicely blend the influence from each node based on the distance away from it. I think this results in a rather curved corners, as can be seen from the purple figure in Fig. 5. The elbows seem rather rubbery, as the deformation parameters in those regions presumably had to be approximated from afar.

Experiments using shapes that have topological differences, e.g. in Fig. 4, seem to be random and do not support the true contributions of the proposed method. As the constraints and losses are designed to match corresponding shapes there is not much point in applying the method to shapes where only semantic correspondence exist, despite seemingly providing robust correspondence where available.

The detail about the order of the polynomial basis $p$ used in the experiments seems to be missing.

---

> ### Author Response · Authors · 2022-08-02
> **Response to reviewer DTPf**
>
> 1. > **The node sampling strategy in Section 4.4 seems to imply that the deformation is mainly articulation-based, rather than a fully non-rigid one.**
>
> We understand that our use of SMPL segments during node sampling might have been ambiguous and led to this inference. Please note that, as mentioned in our answer to question 2 in general comment, this sampling strategy is only necessary to avoid the presence of deformation nodes that influence geodesically distant neighborhoods. Moreover, as we demonstrated in the Sec 3.4, Suppl, simpler and purely geometric strategies lead to similar results. In general, our sampling strategy is geared towards fully non-rigid deformations and is not fundamentally articulation-based. We invite you to consider Sec 5.3 and Fig 4 in the main draft, which illustrates results on man-made shapes. We will be happy to include more explicit results on non-articulated (e.g., organic) shapes and make this point more clear in the revised version.
>
> 2. > **I was actually curious about why this segmentation-based sampling is required**
>
> We believe this is also answered in the general comment (2nd and 3rd question) . We hope our added ablation (Sec 3.4, Suppl) sheds more light on when and where it is necessary. We will be happy to answer if there are additional questions/concerns.
>
> 3. > **Experiments using shapes that have topological differences, e.g. in Fig. 4, seem to be random**
>
> Purpose of experiments in Sec 5.3, main draft (Segmentation transfer) and Sec 5, Suppl (Implicit shape correspondence) are geared towards showing that our representation of the deformation field can handle shapes undergoing topological changes, which might not always be clear for methods based on template deformation.
>
> In addition, they are particularly interesting because in the segmentation transfer experiment, our template is a point cloud rather than a mesh (in contrast to previous two experiments - Sec 5.1, 5.2 of main draft). In the implicit shape correspondence, we show that approximation can be performed over the entire domain (unit cube) by using nodes with variable radii.
>
> 4. > **The detail about the order of the polynomial basis**
>
> We use a polynomial basis of n=1, i.e, $p = [1,x,y,z]^T$. This detail is provided in the supplementary (see L:23).
>
> 5. > **What kind of poses would be difficult to be used as the template?**
>
> From our ablation study on the template pose (Fig 8, Suppl), the difference in performance across different template poses is minimal. The pose of the template is not a strong constraint, as long as sampling of nodes is appropriate, i.e. making sure that nodes do not interfere with points that are far in the geodesic sense. (Sec 3.4, Suppl).
>
> 6. > **Why the data-heavy baselines such as 3D-CODED cannot achieve high correspondence accuracy compared to the proposal.**
>
> We hope this is answered in the first question of general comment. If there is an ambiguity in our reasoning, we will be happy to clarify further.
>
> 7. > **I would like some clarification regarding the test phase.**
>
> $\mathcal{D}_X$ and $\mathcal{D}_Y$ indeed are Template $\mathcal{T}$ deformed by our pipeline to match shapes $\mathcal{X}$ and $\mathcal{Y}$ respectively. Therefore they share the correspondence by ordering, i.e the $i^{th}$ vertex in $\mathcal{D}_X$ and $\mathcal{D}_Y$ are in correspondence. Since deformed templates must match respective shapes $\mathcal{X}$ and $\mathcal{Y}$, nearest neighbor search between $\mathcal{D}_X$ and $\mathcal{X}$ (analogously $\mathcal{D}_Y$ and $\mathcal{Y}$) must be in correspondence. Thank you for pointing this out, we have rectified in the revised version. Hope this provides better clarity.
>
> 8. > **What would happen if the nodes were more evenly distributed, without removing any from near the segment boundaries?**
>
> We believe this is also answered in the general comment and Sec 3.4, Suppl. Short answer - uniform sampling is sensitive towards template poses whereas our prescribed sampling is more robust. Please note that for Sec 5.2, 5.3 - main draft and Sec 5 - Suppl, we do not remove nodes from segment boundaries.
>
> 9. > **involving human figures, the head does not seem to be turning.**
>
> This is purely a rendering choice. We orient all shapes to face the camera at the time of rendering figures. At training/inference time, shapes have arbitrary orientation along Y-axis (c.f last sentence, Section 5.1 of main draft).
>
> 10. > **what would happen for shapes whose parts are difficult to determine (e.g. human organs, etc.)**
>
> To the best of our knowledge, there are no benchmarks consisting of human organs with dense ground truth to perform a quantitative analysis. However, we agree with you that demonstration on such a dataset will be useful for a broader audience. Therefore, we will add qualitative results pertaining to unarticulated non-rigid shapes in our final version of the draft.

---

> > ### Comment · Reviewer_DTPf · 2022-08-09
> > **Thank you for the responses!**
> >
> > I would like to thank the authors for taking the time to clarify my questions.
> > I am really happy that most of the questions have been addressed, but still have some elements that are unclear or concerning:
> >
> > regarding my points and the authors' response to 1,8,10:
> > I am happy if this paper were written to address shape articulation (or shape deformation with clear parts). But the term "non-rigid" to me includes shapes that have more freedom regarding form (including human organs, as I've mentioned in my review) , and the experiments in the paper (as well as the newly added ones) does not truly address such domains. Therefore I would be happier if the authors changed the title slightly to show that the domain is rather targeted to shapes that have relatively clear partwise correspondence between them. I suspect that in cases where parts are difficult to determine, it may be unwise to leave segment boundaries unsampled...
> >
> > In line with the previous point, response to 3:
> > the authors mention that
> > >our representation of the deformation field can handle shapes undergoing topological changes, which might not always be clear for methods based on template deformation.
> >
> > But this isn't too convincing in relation to my prior concern. Fig. 4 and Fig. 12 of the supplementary is provided as evidence, but clear "correspondence" is hard to define when undergoing topological change in my opinion. In Fig. 4, the airplane seems to have one engine missing. Although better contained than the other prior methods, shouldn't this in reality (or to our senses) be non existent? (a non-existent engine should not correspond to anywhere...). The same goes for the sofa and the chair in Fig. 12. Hence my claim that the domain of the proposed method should be limited to shapes with existing partwise correspondence, which would make an excellent paper.
> >
> > The response to 9:
> > I still don't understand after the explanation. Despite the rendering preference, as the head is one segment on its own, shouldn't it be aligned to the corresponding head pose?
> >
> > Thank you!
> > Best regards.

---

> > > ### Author Response · Authors · 2022-08-09
> > > **Addressing reviewer DTPf's concerns**
> > >
> > > Thank you for your suggestions. The definition of "non-rigid objects" could encompass a broader range of object categories. To address them, we are willing to make the necessary clarifications for the final version as follows,
> > >
> > > 1) We will clarify through our text (Abstract, Intro, etc..) that generalizability of non-rigid shapes are ones with clear part-wise correspondence. Unfortunately, while we would be happy to, we do not have the permission to change the title.
> > >
> > > 2) We will also elucidate in Sec 5.3 (main draft) and Sec 5 (Suppl) that handling topological difference is *not* a characteristic of our approach while given limited training data, our reduced representation empirically performs better baseline which uses pointwise MLPs. For a future work that is built to explicitly handle partiality, we believe our representation can be more useful than point-wise MLPs and respective experiments are a proof-of-concept.
> > >
> > > 3) We will add qualitative correspondence results from scans of human hearts [1] to show a broader applicability of our method.
> > >
> > >
> > > > Despite the rendering preference, as the head is one segment on its own, shouldn't it be aligned to the corresponding head pose?
> > >
> > > Finding the correct orientation of the head is arguably one of the most challenging aspects of registering a human body, since different rotations do not produce significant differences in geometric features. However, we highlight that in Fig 2 and Fig 6 of Suppl, the poses of head are distinct w.r.t the trunk showing that the model has learned some structural relations from the data. For the final version, in the Suppl., we will add qualitative examples corresponding to SHREC'19 (Tab1, Col4, main draft) with different head orientations.
> > >
> > >
> > > References
> > >
> > > [1] Strocchi et.al PLOS Journal, 2020, https://doi.org/10.1371/journal.pone.0235145

---

### Official Review · Reviewer_6kov · 2022-07-11

**Rating:** 5
**Confidence:** 4
**Soundness:** 3 good
**Presentation:** 3 good
**Contribution:** 3 good

**Summary:**

This paper presents an efficient approximation of deformation fields using nodes plus local weighted least-square fitting. The proposed representation enables easy computation of Jacobian and thus facilitates first-order regularizations. Experiments on multiple tasks show that it performs well compared to recent existing methods.

**Questions:**

Questions:
- In the node sampling in Section 4.4 it says the method "exclude(s) a node that exerts its influence in semantically different regions" (line 223), does it mean that semantic segmentations/annotations are implicitly used? -- although it seems to be a minor issue as the sampling may not be the core of the proposed method, using this information still sounds improper as they can be strong hints to correspondences (or in other words, if you already have access to this information, why not also using them in the correspondence computation...). I wonder how the method works if the node samplings involve no semantics? Or correct me with how "semantically different regions" are defined -- my understanding is that the segmentations are from SMPL[30] ground-truth annotations.

Suggestions:
- In Section 3.2, it can be made clearer which contributions are from prior works and which are proposed in this paper. For instance, the Moving Least Squares are from [16], but for the following shape representations, are they simply existing formulations, or are there any adaptations to the specific settings here?

Other minor suggestions:
- (line 3) "Differently from" -> "Different from"
- The plot in Figure 5 left is too small. It's a bit hard to read the caption and legends.

**Limitations:**

The authors have addressed the limitations of their work.

**Strengths And Weaknesses:**

Strengths:
- The paper introduces a deformation field representation for more efficient learning of shape correspondences. It also enables easy computations of multiple regularization losses based on its accessibility to the Jacobian.
- Good experimental results show the effectiveness of the proposed method compared to prior works. The authors also presented multiple applications such as interpolation and unsupervised segmentation.

Weaknesses:
- I feel the point of preserving "high-frequency components" (claimed in line 32-37) is not well addressed in the experiments. Apart from the generic tasks, specific experiments demonstrating the method's superiority on this can make the argument stronger.

---

> ### Author Response · Authors · 2022-08-02
> **Response to reviewer 6kov**
>
> 1. >**Point of preserving "high-frequency components" (claimed in line 32-37) is not well addressed in the experiments.**
>
> We agree with the reviewer that terminology of high and low frequency might be confusing.Therefore, we have modified “frequency” with the phrase  “details”.
> However, we believe this claim is empirically addressed through the question of "*why our approach needs fewer refinement steps*” in comparison to baseline. Please see Sec 3.2 and Fig 6 of Suppl where our approach effectively reconstructs finer details that the baseline (3D-CODED) fails to. Also please see Fig 5 of Suppl where we show the individual contributions of regularization in achieving detail-preserving reconstruction.
>
> 2. >**Relation between semantic information and hints to correspondence**
>
> We hope that our answer in the general comment clarifies the motivation behind using segmentation for sampling. Please note that the segmentation information was used only on the template shape and only in the pre-processing step. The purpose of segmentation is to deduce regions where a node cannot be positioned. The position of nodes themselves do not contain any information pertaining to correspondence. In fact across all our experiments, nodes need not necessarily be on the surface of the template (see L:222). Moreover, this information was not used for any other shapes, neither at pre-processing nor at training nor at inference time. For a given target shape, we estimate deformation values at nodes solely from its coordinates in the Euclidean space.
>
> In addition, this semantic information cannot be leveraged at the inference time for computing correspondence without the same process being repeated on a target shape. Since all benchmarks that we used in quantitative evaluations are extraneous to SMPL templates, using SMPL information is inapplicable.
>
> If there is still ambiguity in our explanation, we will be happy to clarify further questions.
>
> 3. > **how "semantically different regions" are defined**
>
> Semantically different regions are defined by segmenting the template mesh in order to prevent a node from influencing points across different segments. Please note, SMPL annotations is not the unique way of segmenting the template whereas it is one possible option (general comment, question 2). As shown in Sec 3.4 of Suppl, an alternative purely geometric segmentation strategy[1] achieves the same performance without requiring ground truth or being restrictive to SMPL templates.
>
> 4. > **how the method works if the node samplings involve no semantics?**
>
> We hope this is clarified through our general comment and Sec 3.4 of Suppl. In short, a uniform sampling strategy is sensitive to the pose of the template while including semantic information (not SMPL in particular) makes the performance robust to changes in pose.
>
> 5. > **Contribution from prior work In Section 3.2.**
>
> We adapt the compact weighting function which was used by Adams et.al [2] in our representation. We have clarified this in our revised main draft.
>
> 6. > **The plot in Figure 5 left is too small.**
>
> We apologize for this. We have updated the figure with improved legibility of qualitative results.
>
>
>
> ### References
>
>
> [1] Raif M. Rustamov, "Laplace-Beltrami Eigenfunctions for Deformation Invariant Shape Representation", SGP 2007
>
> [2] Adams et.al. "Meshless Modeling of Deformable Shapes and their Motion", SCA 2007

---

> > ### Comment · Reviewer_6kov · 2022-08-09
> > **Reply**
> >
> > Dear authors,
> >
> > Thanks for your reply! I think I am convinced by these answers.
> >  - Yes, 'preserving details' sounds better than 'preserving high-frequency components' to me -- especially under the context of correspondence, 'preserving high-frequency components' can usually mean the method's properties on the Laplacian (or other operators') eigenfunctions due to the prevalence of spectral methods. The empirical results also make sense to me.
> >  - As the semantic segmentations are not used anywhere other than the template shape, it sounds very reasonable to me then.
> >
> > Also thanks for all the revisions on the paper and supplementary material.

---

### Official Review · Reviewer_u8B8 · 2022-07-12

**Rating:** 7
**Confidence:** 3
**Soundness:** 3 good
**Presentation:** 3 good
**Contribution:** 3 good

**Summary:**

The paper describes a method to compute correspondences from shapes with application for shape registration, shape interpolation and unsupervised Segmentation Transfer. The authors use an implicit (meshless) shape reconstruction based on the moving least squares as a shape representation. An autoencoder provides a way to create a compact representation of shapes which is then used to solve correspondence problems.

**Questions:**

- The choice of the control points seems to be essential to obtain a compact representation that captures the important details. Explain how to properly choose the number K of those points ? Why not make it as large as possible ? How the local radius r_i are chosen ?

- It is not clear where the section "Extending sparse to dense Correspondence" and Eq(12) is used in the result section. Make this more explicit  or remove the section.

- In the test time refinement, it is difficult to understand what is being done without reading the 3D-CODED paper. Improve this section. Why is the optimization taking place only for Z_y and not for Z_X ? This breaks the symmetry between the 2 shapes.


**Ethics Review Area:**

["I don’t know"]

**Limitations:**

- The authors have cited 2 limitations (non diffeomorphic deformations + optimization of control points) which are relevant

**Strengths And Weaknesses:**

Strengths :
i) Novel shape representation for shape encoding inspired from meshless representations.
ii) extensive experiments showing the ability of the proposed approach to solve  shape registration, shape interpolation and unsupervised segmentation transfer problems.
iii) The paper is fairly well written with a lot of implementation details in the supplementary material

weaknesses :
i) the considered problem assumes that the correspondences between shapes are known at training time
ii) Only displacement fields are estimated that may lead to non diffeomorphic deformations. This may be a limitation
iii) the main concept of the paper is borrowed from 3D-CODED but the shape representation is original

---

> ### Author Response · Authors · 2022-08-02
> **Response to reviewer u8B8**
>
> 1. > **Choosing number of nodes and its radius**
>
> The number of nodes and their influence (radius) are important parameters in our reduced representation. We seek a representation that is compact and preserves local characteristics of deformation (L:126, main draft), i.e. fewer nodes and a smaller radius. Unfortunately, they both cannot be true simultaneously as it could potentially lead to a singular moment matrix (L:133-134, main draft). Therefore, we first make a choice on the compactness by choosing the radius of each node to be 1/10th of shape diameter. Then, we select points from candidates (see Fig 2(b), 2(c), Suppl) until the non-singularity condition for the moment matrix is satisfied. Since this is a hyper-parameter based on our choice, we have added an ablation (Sec 3.3, Suppl) to compare compactness and influence.
>
> 2. > **Why not make the number of nodes/radius as large as possible ?**
>
> An increase in the number of nodes implies more learnable deformation parameters, which necessitates an increased training data (see our general comment, point 1). Also, a larger radius would influence points on the template that are far from each other in a geodesic sense, making it difficult to capture the local characteristics of deformation. (our general comment, point 2)
>
> Indeed, the two explanations above corroborates with the observed results in ablation study (Sec 3.3, Suppl).
>
> 3. > **Optimization taking place only for Z_y and not for Z_X ?**
>
> The optimization takes place for both X and Y conversely to what equation (11) was in the original submission. We acknowledge this mistake and we have rectified it in the revised version.
>
> 4. > **Where "Extending sparse to dense Correspondence" and Eq(12) is used in the result section?**
>
> We use this in Sec 5.2, main draft for the task of shape registration, only on training shapes. Since the SHREC’20 benchmark consists of key-point correspondence annotation and the data-driven baselines we consider are not built to handle such sparse ground truths, for a fair comparison, we train all methods (including ours) using this dense correspondence. We will make this clear in our final manuscript. Note, this extension to dense correspondence is not performed on test shapes for evaluation.

---

### Author Response · Authors · 2022-08-02
**General Comment: Response to all reviewers**

We thank all reviewers for their insightful comments and questions. Before addressing individual concerns in separate comments, we begin by stating the revisions made to our submission and address common questions among reviewers in the rebuttal.

## Revisions
We have revised our main draft only with modifications that enhance clarity and do not warrant additional space. Additional changes promised in the rebuttal will be incorporated in the final version due to the scope of an extra page allowance upon acceptance. All references made in the rebuttal are with respect to revised drafts. Summary of revision is as follows,

### Main Draft

1. Revised inference section to better clarify the refinement and correspondence computation steps.
2. Modified Figure 5 with improved legibility of quantitative results.
3. Replaced high/low frequency with coarser/finer details.
4. Added a citation to section 3.2 to clarify an adaptation from prior work.


### Supplementary
1. Added an ablation study (Sec 3.3) to analyze the effect of number of nodes and its influence (radius).
2. Added an ablation study (Sec 3.4) to compare node sampling strategies.

## Response to common questions
We address common points raised by multiple reviewers below,

1. > **Why our pipeline requires fewer training samples to achieve optimal performance?**

We attribute three main reasons for requiring fewer training samples.

1.  Using a compact set of parameters (deformation nodes) can be thought of as providing a reduced and expressive basis for the underlying deformation field. Different from competing baselines, we do not *learn* local/finer components, instead, from the learned deformation parameters at nodes (global/coarser form), the local components are computed in closed-form (c.f Eqn 3, main draft).

2. We apply efficient first-order regularization, by imposing approximate volume preservation and promoting as-rigid-as-possible deformations. This leads to well-structured smooth deformation fields (Fig 5, Tab 5 - Suppl), without requiring extensive training data.

3. We regularize intermediate shapes between two shapes (c.f Eqn 10, main draft) thereby forgoing the need for explicit supervision to enforce plausible deformations of intermediate shapes. Fig 5, main draft illustrates how the deformation of intermediate shapes is naturally smooth. This regularizing all intermediate shapes is computationally feasible (see Tab 4, Suppl for timing) only due to a closed form expression of the Jacobian (c.f Eqn 4, main draft and Sec 1.2, Suppl), which is not the case with other baselines.

We agree it is essential to be clear with our reasoning behind reduced training effort. To that end, we will summarize this reasoning in the final version.

2. > **Why is a segmentation based sampling required?**

We define the deformation nodes over the template shape and the deformation parameters at those nodes are learned for a given (target) shape by our network. However, depending on *where* each node is positioned, the approximation (c.f Eqn 3, main draft) influences the deformation field in its vicinity. Typical uniform sampling might lead to a node jointly approximating the deformation field at two points in the template that are geodesically far (see Fig 7(b), Suppl). To prevent placing nodes at such positions, we use semantic information. However, our use of SMPL’s segmentation is only “*one of the possible choices*”. We show a more generic alternative in Sec 3.4 of Suppl that is equally efficient and does not rely on any external/ground truth annotation. Also, with uniform sampling, a change in pose of the template worsens the accuracy whereas the difference in performance is negligible when using segmentation sampling (Tab 7, Suppl). Therefore, a segmentation based sampling is more robust to changes in template pose without entailing a particular way (e.g: SMPL annotation) to segment the template. We will better clarify this in the final version.

3. > **When is a segmentation based sampling required?**

When the influence of deformation nodes can encompass geodesically distant points (due to the pose of the template), then a segmentation based sampling would be an ideal choice. E.g: for a template in A-pose, segmentation sampling is seemingly effective. Alternatively, for a template in T-pose, a node influencing geodesically distant points is unlikely. In such cases, a uniform sampling shows comparable performance. Please see Sec 3.4 and Tab 7 of Suppl for more detail.

4. > **What happens without such sampling?**

We report three experiments where no rejection sampling was applied. Namely for shape-registration (Sec 5.2, main draft), segmentation transfer (Sec 5.3, main draft) and implicit matching (Sec 5, Suppl). This is because the template is in T-pose for shape-registration while the template in the latter two experiments are point clouds with no connectivity.

---

### Author Response · Authors · 2022-08-09
**Gentle Reminder: Nearing end of author-reviewer discussion period**

Dear Reviewers,

As we near the end of author-reviewer discussion period, we would like to mention again that we will be happy to answer any further questions or elucidate on our previous answers. Given the efficacy of our proposed novel reduced representation, its strong empirical performance and now with better clarity on sampling strategies, we believe that our contributions will be of considerable interest to the NeurIPS audience. We have addressed all concerns raised in the initial review to the best of our knowledge, but will be happy to provide further information or answer any remaining questions you might have. Thank you for your thoughtful and detailed feedback.

---

### Meta-Review · Area_Chair_3MUs · 2022-08-25

**Recommendation:** Accept
**Confidence:** Certain

**Metareview:**

This paper presents a creative approach to shape registration that incorporates a new parameterization of the deformation field.  All authors (and the AC) agree the results are convincing and that the method presents novel and interesting ideas.

The only (borderline) negative review by reviewer 86v4 seems to be well-addressed by the rebuttal and new ablation experiments; although reviewer 86v4 did not engage during the rebuttal discussion, the AC checked these results and found them reasonable.

Hence, a recommendation of "accept" is suitable here.  The final version of the paper should be sure to incorporate any new experimental results that appear in the rebuttal discussion.

**Award:**

No

---

### Decision · Program_Chairs · 2022-09-14

Accept